# Mechanism of early light signaling by the carboxy-terminal output module of *Arabidopsis* phytochrome B

Yongjian Qiu[1], Elise K. Pasoreck[2], Amit K. Reddy[2], Akira Nagatani[3], Wenxiu Ma[4], Joanne Chory[5] & Meng Chen [1]

Plant phytochromes are thought to transduce light signals by mediating the degradation of phytochrome-interacting transcription factors (PIFs) through the N-terminal photosensory module, while the C-terminal module, including a histidine kinase-related domain (HKRD), does not participate in signaling. Here we show that the C-terminal module of *Arabidopsis* phytochrome B (PHYB) is sufficient to mediate the degradation of PIF3 specifically and to activate photosynthetic genes in the dark. The HKRD is a dimerization domain for PHYB homo and heterodimerization. A D1040V mutation, which disrupts the dimerization of HKRD and the interaction between C-terminal module and PIF3, abrogates PHYB nuclear accumulation, photobody biogenesis, and PIF3 degradation. By contrast, disrupting the interaction between PIF3 and PHYB's N-terminal module has little effect on PIF3 degradation. Together, this study demonstrates that the dimeric form of the C-terminal module plays important signaling roles by targeting PHYB to subnuclear photobodies and interacting with PIF3 to trigger its degradation.

[1] Department of Botany and Plant Sciences, Institute for Integrative Genome Biology, University of California, Riverside, CA 92521, USA. [2] Department of Biology, Duke University, Durham, NC 27708, USA. [3] Department of Botany, Graduate School of Science, Kyoto University, Kyoto 606-8502, Japan. [4] Department of Statistics, University of California, Riverside, CA 92521, USA. [5] Howard Hughes Medical Institute, Plant Biology Laboratory, The Salk Institute for Biological Studies, La Jolla, CA 92037, USA. Correspondence and requests for materials should be addressed to M.C. (email: meng.chen@ucr.edu)

Phytochromes (PHYs) are evolutionarily conserved photo-receptors in bacteria[1], fungi[2], algae, and plants[3–5]. In plants, PHYs are red (R) and far-red (FR) photoreceptors that can be photoconverted between two relatively stable forms: the R light-absorbing inactive Pr form and the FR light-absorbing active Pfr form[6, 7]. PHYs regulate almost all aspects of plant develop-ment and growth, including germination, de-etiolation, shade avoidance, plant defense, floral induction, and senescence[8, 9]. The importance of PHYs in plant development and growth is best exemplified in *Arabidopsis* de-etiolation. When young seedlings emerge from the ground and first encounter light, photoactiva-tion of PHYs triggers a dramatic developmental transition from etiolation, a dark-grown developmental program, to photo-morphogenesis, a light-grown developmental program that restricts hypocotyl growth and promotes chloroplast biogenesis and photoautotrophic growth[10]. These diverse photo morphological responses in *Arabidopsis* are mediated by five PHYs, PHYA-E[11], among which PHYB plays a prominent role[12].

PHYs trigger photomorphogenesis by reprogramming the nuclear genome[13, 14]. One of the earliest light responses at the cellular level is the translocation of photoactivated PHYs from the cytoplasm to the nucleus[15–17], where PHYs interact directly with a group of nodal basic helix-loop-helix transcriptional regulators —the phytochrome-interacting factors (PIFs)—and regulate their stability and activity[18–22]. The PIFs belong to subfamily 15 of the bHLH protein superfamily in *Arabidopsis* and include eight members: PIF1, PIF3-8, and PIL1 (PIF3-Like1)[23, 24]. In general, PIFs play antagonistic roles in photomorphogenesis, including promoting hypocotyl elongation and repressing chloroplast bio-genesis, with different PIFs performing overlapping and distinct roles[25, 26]. PIF1, PIF3, PIF4, PIF5, and PIF7 promote hypocotyl growth by activating growth-relevant genes, such as genes involved in the biosynthesis and signaling of the plant growth hormone auxin[25–28]. PIF1, PIF3, and PIF5 inhibit chloroplast biogenesis by repressing nuclear-encoded photosynthetic genes[26, 29–32]. Most PIFs accumulate to high levels in dark-grown seed-lings, and their protein levels are rapidly dampened in the light by PHYs[18–23, 33]. Our understanding of PIF regulation in early PHY signaling came first from extensive studies of the founding member of the PIFs—PIF3[34]. PIF3 interacts preferentially with the active Pfr forms of PHYA and PHYB[34, 35]. The PHY-PIF3 interaction promotes phosphorylation and subsequent degrada-tion of PIF3 in the light by the ubiquitin-proteasome pathway[18]. PIF3 degradation is carried out by the Cullin3-based E3 ubiquitin ligases containing the substrate recognition proteins LRB1-3 (light-response broad-complex/Tramtrack/Bric-a-brac)[36] and requires a PHY- and PIF3-interacting transcriptional coactivitor, HEMERA (HMR)[33, 37–39]. Because PIF3 is recruited to PHYB-containing subnuclear photosensory domains named photobodies during the dark-to-light transition prior to its degradation[18, 40], it was proposed that PIF3 degradation occurs at photobodies. This hypothesis is supported by a tight correlation between photobody disassembly and PIF3 accumulation during the light-to-dark transition[41] and by the genetic evidence that the *hmr* mutant—which is defective in photobody biogenesis—also fails to degrade PIF3 in R light[33, 37, 38]. However, how the PHYB-PIF3 interaction induces PIF3 degradation and transcriptional regulation of its target genes is still not fully understood.

A major task in understanding early PHY signaling is to identify and dissect the functional roles of PHY's individual conserved domains in the early PHY phototransduction events of nuclear accumulation, photobody localization, as well as PIF interaction and degradation. The prototypical plant PHY is a homodimer, each monomer contains an N-terminal photo-sensory module and a C-terminal output module[6, 7]. The N-terminal photosensory module consists of four subdomains—

an N-terminal extension that is essential to stabilize the Pfr form and can be negatively regulated by phosphorylation[42–44], a PAS (period-Arnt-single-minded) domain of unknown function, a GAF (cGMP photosphodiesterase/adenylate cyclase/FhlA) domain that binds a bilin chromophore, and a PHY (phyto-chrome-specific) domain that stabilizes the photoactivated Pfr conformer[6, 45]. Structural studies of the PAS-GAF-PHY domains of bacteriophytochromes[46–48] and *Arabidopsis* PHYB[44] show that the PHY domain contributes a hairpin loop protrusion or "ton-gue," which is in close contact with the bilin-binding pocket of GAF and undergoes a β-stranded to helical conformational conversion during the Pr-to-Pfr photoactivation. In addition, at the PAS-GAF interface, there is an unusual figure-eight knot called the "light-sensing knot lasso"[44, 46, 47, 49], which interacts directly with PIFs[50, 51]. This interaction is considered as a trigger for PIF3 degradation[18, 51, 52].

The C-terminal output module of PHYs contains two tandem PAS domains named PRD (PAS-repeat domain) and a histidine kinase-related domain (HKRD). Previous studies have shown that the C-terminal module of PHYB forms a dimer[53, 54] and the HKRD in both PHYA and PHYB was postulated as a dimeriza-tion domain[54–57]. In PHYB, the C-terminal module plays important roles in nuclear accumulation and photobody locali-zation[17, 53, 58]. The PRD has been demonstrated to mediate PHYB's nuclear localization, and the entire C-terminal domain is required for photobody biogenesis[53, 58]. The HKRD exhibits high sequence similarity to bacterial histidine kinases[59, 60]. In fact, bacterial and fungal PHYs are bona fide histidine kinase sensors, and their histidine kinase domains are the signaling-output domain that relays phototransduction through autopho-sphorylation of an active-site histidine and phosphotransfer to an aspartate in a cognate response regulator[1, 61, 62]. In contrast, PHYs in higher plants lack the conserved active-site histidine and instead have Ser/Thr kinase activity[60]. Interestingly, deleting the majority of the HKRD had only minimum effects on PHYB signaling[63]. Therefore, the prevailing model is that the C-terminal module of PHYB does not participate in signaling output, in particular the HKRD is dispensable[7, 53, 63].

Although the current view of the structure-function relation-ship of plant PHYs has been widely accepted, a number of lines of evidence still support a signaling role for the C-terminal module. For example, many loss-of-function mutations in PHYA and PHYB fall within the C-terminal module[6]. Among the reported *phyB* alleles, the *phyB-18* mutant is unique because it carries a D1040V mutation in the HKRD and is the only missense *phyB* allele identified in the HKRD[63]. More interestingly, although deleting the majority of the HKRD had a minor effect on PHYB activity, the *phyB-18* mutant is severely defective in PHYB-mediated light responses[63]. To further explore possible signaling roles of the C-terminal module of PHYB, we have investigated how D1040V abolishes PHYB signaling in the *phyB-18* mutant. We demonstrate that the entire HKRD is a dimerization domain within the C-terminal module of PHYB and that the D1040V mutation abrogates the dimerization of HKRD and consequently attenuates the early signaling functions of PHYB in nuclear accumulation, photobody localization, interaction with PIF3, and PIF3 degradation. Our results show unexpectedly that the C-terminal module, but not the N-terminal module, of PHYB plays a direct and essential signaling-output role in PIF3 degradation.

## Results

**phyB-18 impairs nuclear accumulation and PIF3 degradation.** To investigate how the D1040V mutation in *phyB-18* abrogates PHYB signaling, we generated transgenic lines in a *phyB* null allele, *phyB-9*, expressing PHYB18 fused with yellow fluorescent

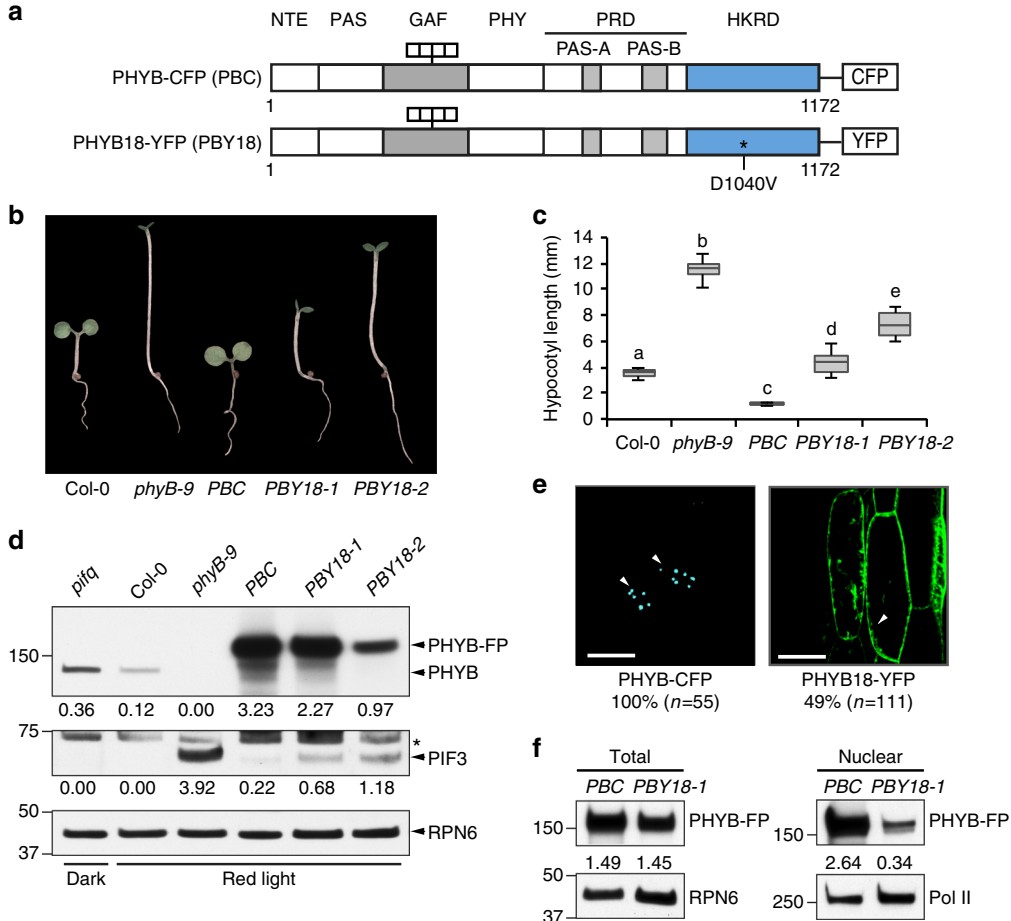

**Fig. 1** *phyB-18* impairs PHYB nuclear accumulation and PIF3 degradation. **a** Schematic illustration of the domain structures of PHYB-CFP (PBC) and PHYB18-YFP (PBY18). The *phyB-18* mutant carries a D1040V mutation in the HKRD. **b** Representative images of 4-day-old Col-0, *phyB-9*, *PBC*, *PBY18-1*, *PBY18-2* seedlings grown in 10 μmol m$^{-2}$ s$^{-1}$ R light. **c** Box and whisker plots of hypocotyl length measurements of the seedlings shown in **b**. The boxes represent from 25th to 75th percentile; the bars equal to the median values. Samples with different letters exhibit statistically significant differences in hypocotyl length (ANOVA, Tukey's HSD, $P < 0.01$, $n > 40$). **d** Immunoblots showing the protein levels of PHYB, PHYB-FP and PIF3 in the indicated lines grown either in the dark or 10 μmol m$^{-2}$ s$^{-1}$ R light. RPN6 was used as a loading control. The relative levels of PHYB, PHYB-FP or PIF3 were normalized to the corresponding levels of RPN6 and are shown below each lane. The asterisk indicates nonspecific bands. **e** Confocal images showing the subcellular localization patterns of PHYB-CFP and PHYB18-YFP in hypocotyl cells of 4-day-old R-light-grown *PBC* or *PBY18-1* seedlings, respectively. White arrowheads indicate the nuclei. Scale bars equal 20 μm for the PHYB-CFP image and 40 μm for the PHYB18-YFP image. PHYB-CFP was localized to subnuclear photobodies in 100% of the cells ($n = 55$), whereas PHYB18-YFP was mainly localized to the cytoplasm and was observable in the nucleus in only 49% of the cells ($n = 111$). **f** Immunoblots showing the total and nuclear fraction of PHYB-CFP and PHYB18-YFP. RPN6 and Pol II were used as loading controls for total and nuclear proteins, respectively. The relative protein levels of PHYB-CFP and PHYB18-YFP were normalized to RPN6 (total) or Pol II (nuclear) and are shown below the blots. PRD PAS-repeat domain, PHY phytochrome domain, GAF cGMP phosphodiesterase/adenylyl cyclase/FhlA domain

protein, PHYB18-YFP (PBY18) (Fig. 1a). The phenotypes of two independent homozygous transgenic lines, *PBY18-1* and *PBY18-2*, were compared with a previously described transgenic line in *phyB-9* expressing the wild-type PHYB fused with cyan fluorescent protein, PHYB-CFP (PBC) (Fig. 1a)[58]. We first examined the activity of PBY18 in mediating photo-inhibition of hypocotyl growth. When grown under monochromatic R light, the *PBC* line rescued the long-hypocotyl phenotype of *phyB-9*. In contrast, neither *PBY18-1* nor *PBY18-2* could rescue *phyB-9* to the degree of *PBC* (Fig. 1b, c). Between *PBY18-1* and *PBY18-2*, *PBY18-1* was considerably shorter than *PBY18-2* (Fig. 1b, c), which is likely due to the higher expression level of PHYB18 in *PBY18-1* (Fig. 1d). Although the level of PHYB18 in *PHY18-1* was comparable to the level of PHYB-CFP in *PBC* (Fig. 1d), *PHY18-1* was more than twice as tall as *PBC* (Fig. 1b, c). These results are consistent with the previously reported long-hypocotyl phenotype of the *phyB-18* mutant[63], confirming that PBY18 is impaired in PHYB signaling.

PHYs restrict hypocotyl growth by promoting the turnover of the master growth regulators, the PIFs[18–21, 25, 26]. We therefore examined whether PBY18 affected the steady-state level of the prototypical PIF—PIF3[18, 34]. In continuous R light, PIF3 is actively degraded in the wild-type Col-0 but accumulates to a high level in *phyB-9* (Fig. 1d)[18, 33]. Interestingly, the defect of *phyB-9* in PIF3 degradation was rescued in *PBC* but not in *PHYB18-1* and *PHYB18-2* (Fig. 1d), indicating that the D1040V mutation attenuates the function of PHYB in PIF3 degradation.

The current model suggests that PIF3 degradation requires photobody biogenesis in the nucleus[18, 33, 40, 41, 64]. Therefore, we tested whether the D1040V mutation affected the subcellular localization of PHYB. Indeed, while PBC was localized to photobodies in 100% of nuclei in R light[58], PBY18 was localized mainly to the cytoplasm and in 49% of the cells, it was only marginally observable in the nucleus (Fig. 1e). To confirm the defect of PBY18 in nuclear accumulation, we measured the

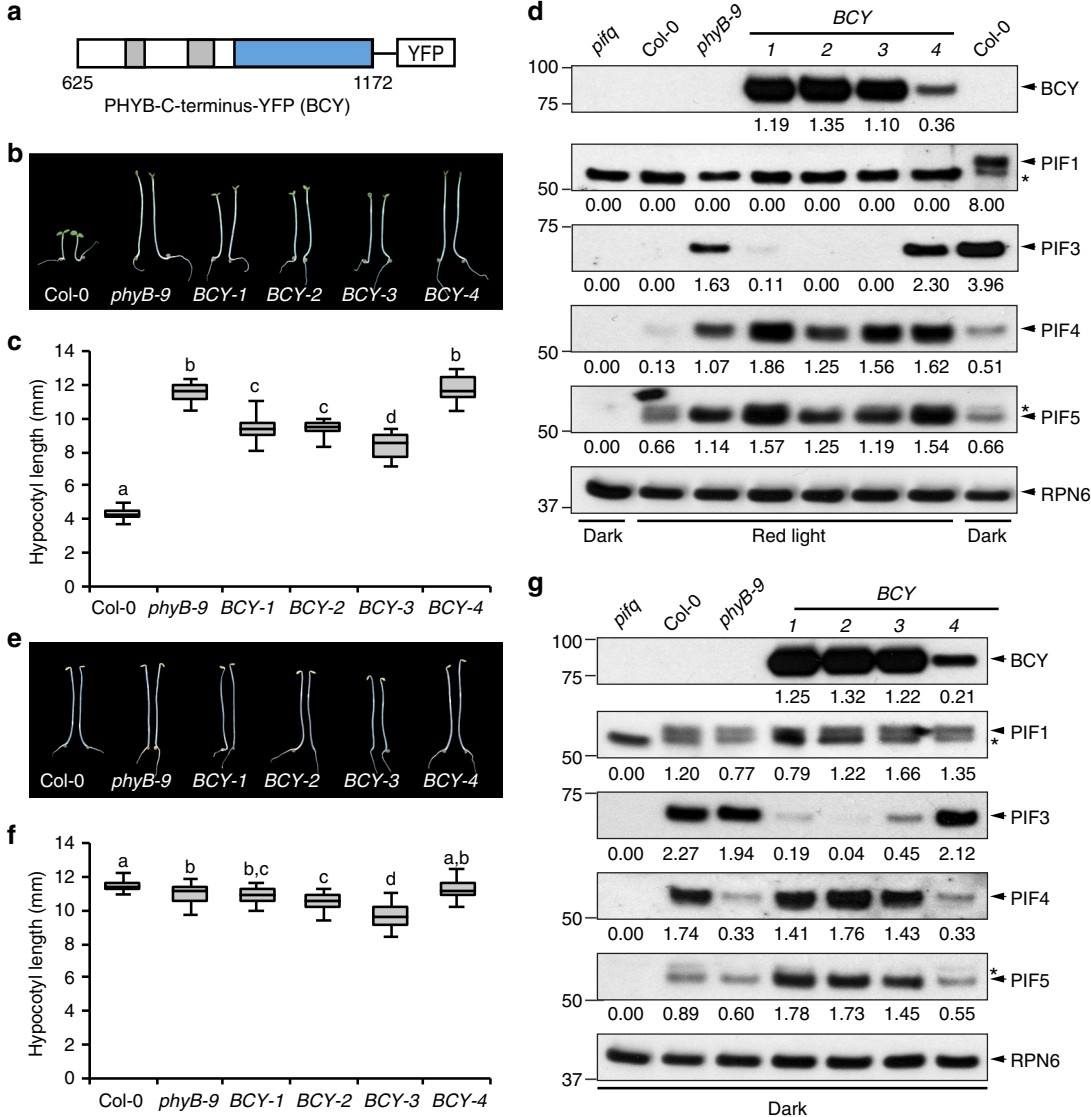

**Fig. 2** The C-terminal module of PHYB is biologically active to regulate hypocotyl growth and mediate PIF3 degradation. **a** Schematic illustration of the domain structures of PHYB-C-terminus-YFP (BCY). **b** Images of 4-day-old Col-0, *phyB-9*, and *BCY-1* to *BCY-4* transgenic lines grown in 10 μmol m$^{-2}$ s$^{-1}$ R light. **c** Hypocotyl measurements of Col-0, *phyB-9*, and *BCY* transgenic lines grown in 10 μmol m$^{-2}$ s$^{-1}$ R light. The boxes represent from 25th to 75th percentile; the bars equal to the median values. Samples with different letters exhibit statistically significant differences in hypocotyl length (ANOVA, Tukey's HSD, $P < 0.01$, $n > 40$). **d** Immunoblots showing the steady-state levels of BCY, PIF1, PIF3, PIF4, and PIF5 in the indicated lines grown in 10 μmol m$^{-2}$ s$^{-1}$ R light. Dark-grown Col-0 and *pifq* samples were used as positive and negative controls for PIFs, and RPN6 was used as a loading control. The relative protein levels were normalized to the corresponding levels of RPN6 and are shown below each lanes. **e** Images of 4-day-old Col-0, *phyB-9*, and *BCY-1* to *BCY-4* transgenic lines grown in the dark. **f** Hypocotyl measurements of Col-0, *phyB-9*, and *BCY* transgenic lines grown in the dark. The boxes represent from 25th to 75th percentile; the bars equal to the median values. Samples with different letters exhibit statistically significant differences in hypocotyl length (ANOVA, Tukey's HSD, $P < 0.01$, $n > 40$). **g** Immunoblots showing the steady-state levels of BCY, PIF1, PIF3, PIF4, and PIF5 in the indicated lines grown in the dark. Dark-grown Col-0 and *pifq* samples were used as positive and negative controls, respectively, for PIFs. RPN6 was used as a loading control. The relative protein levels were normalized to the corresponding levels of RPN6 and are shown below each lanes

nuclear fractions of PHYB in light-grown *PBC* and *PBY18-1* seedlings. Despite the similar amounts of total PHYB-CFP and PHYB18-YFP in *PBC* and *PBY18-1*, respectively, the nuclear fraction of PBY18-YFP was almost 90% less than that of PHYB-CFP (Fig. 1f). These results demonstrate that *phyB-18* is defective in PHYB nuclear accumulation and indicate that the HKRD plays a critical yet unknown role in PHYB nuclear accumulation. The correlation between the defects in PHYB18's nuclear accumulation and PIF3 degradation provides new evidence supporting the notion that the PHYB-mediated PIF3 degradation occurs in the nucleus[23].

**The C-terminal module of PHYB can mediate PIF3 degradation.** The C-terminal module of PHYB is both required and sufficient for mediating PHYB's nuclear accumulation and photobody localization[53, 58]. Therefore, we initially wanted to test whether the D1040V mutation impairs the nuclear- and photobody-targeting of the C-terminal module of PHYB. To that end, we generated transgenic lines in *phyB-9* expressing the C-terminal module of either the wild-type PHYB or PHYB18 fused to YFP[53]. We characterized four independent transgenic lines expressing the C-terminal module of PHYB fused with YFP (*BCY*) (Fig. 2a). Unexpectedly, these experiments revealed that

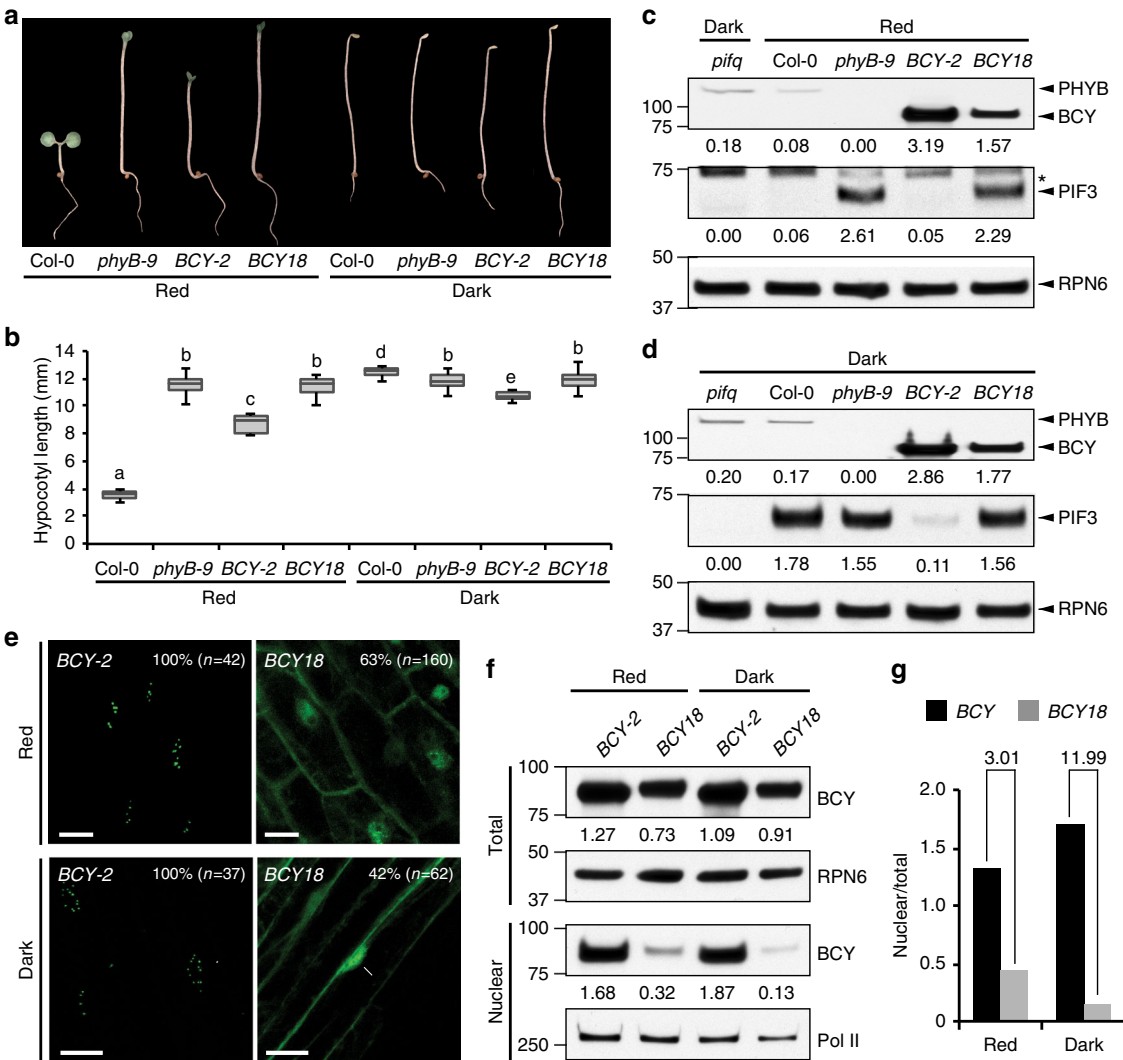

**Fig. 3** D1040V attenuates the signaling and subcellular-targeting activities of the C-terminal module of PHYB. **a** Images of 4-day-old Col-0, *phyB-9*, *BCY-2*, *BCY18* seedlings grown either in the dark or 10 µmol m$^{-2}$ s$^{-1}$ R light. **b** Box and whisker plots of hypocotyl length measurements of the respective seedlings shown in **a**. The boxes represent from 25th to 75th percentile; the bars equal to the median values. Samples with different letters show statistically significant differences in hypocotyl length (ANOVA, Tukey's HSD, $P < 0.05$, $n > 40$). **c** Immunoblot analysis of the levels of PHYB and PIF3 in 4-day-old Col-0, *phyB-9*, *BCY-2*, *BCY18* seedlings grown in 10 µmol m$^{-2}$ s$^{-1}$ R light. A sample from 4-day-old dark-grown *pifq* seedlings was used as a negative control for the PIF3 immunoblot. RPN6 was used as a loading control. The asterisk indicates nonspecific bands in the PIF3 immunoblot. The relative levels of PHYB and BCY were normalized against the corresponding levels of RPN6 and are shown below the blots. **d** Immunoblot analysis of the levels of PHYB and PIF3 in 4-day-old Col-0, *phyB-9*, *BCY-2*, *BCY18* seedlings grown in the dark. RPN6 was used as a loading control. The relative levels of phyB and BCY were normalized against the corresponding levels of RPN6 and are shown below the blots. **e** Representative confocal images showing the subcellular localization patterns of BCY and BCY18 in hypocotyl epidermal cells of 4-day-old seedlings grown either in 10 µmol m$^{-2}$ s$^{-1}$ R light or darkness. The percentage of nuclei showing nuclear localization of BCY or BCY18 in each transgenic line is shown. **f** Immunoblots showing the total and nuclear fraction of BCY and BCY18 in *BCY-2* and *BCY18* lines grown either in 10 µmol m$^{-2}$ s$^{-1}$ R light or darkness. RPN6 and Pol II were used as loading controls for the total and nuclear fraction, respectively. The relative protein levels of BCY and BCY18 were normalized to RPN6 (total) or Pol II (nuclear) and are shown below the blots. **g** BCY18 is defective in nuclear accumulation. The relative nuclear fractions of BCY and BCY18 were estimated based on the ratio between the relative amounts of nuclear and total proteins obtained in **c**. The relative nuclear fractions of BCY18 is 3.01 and 11.99 fold less than those of BCY in R light and darkness, respectively

the C-terminal module of PHYB is biologically active in mediating PHYB signaling. First, although BCY did not fully rescue the long-hypocotyl phenotype of *phyB-9*, three *BCY* lines, *BCY-1* to *BCY-3*, were slightly but significantly shorter than *phyB-9* in R light (Fig. 2b, c). The long-hypocotyl phenotype of the *BCY* lines correlated with the *BCY* expression levels. For example, *BCY-4* expressed the least amount of BCY (Fig. 2d) and had the longest hypocotyl (Fig. 2b, c). More interestingly, the defect of *phyB-9* in PIF3 degradation in R light was largely rescued in *BCY-1* to *BCY-3* (Fig. 2d). The level of PIF3 in *BCY-4* was similar to that of *phyB-9* (Fig. 2d), indicating that the steady-state level of PIF3 is dependent on the amount of BCY.

Because BCY is constitutively localized to the nucleus independent of light[53], we asked whether BCY could confer its signaling activities in the dark. Indeed, the levels of PIF3 in *BCY-1* to *-3* were 4- to 48-fold less than that in *phyB-9* in the dark (Fig. 2g). These results demonstrate that PHYB's C-terminal module is sufficient to mediate PIF3 degradation.

Although PIF3 was largely reduced *BCY-1 to -3* seedlings, the *BCY* lines showed minimum hypocotyl phenotype in the dark—

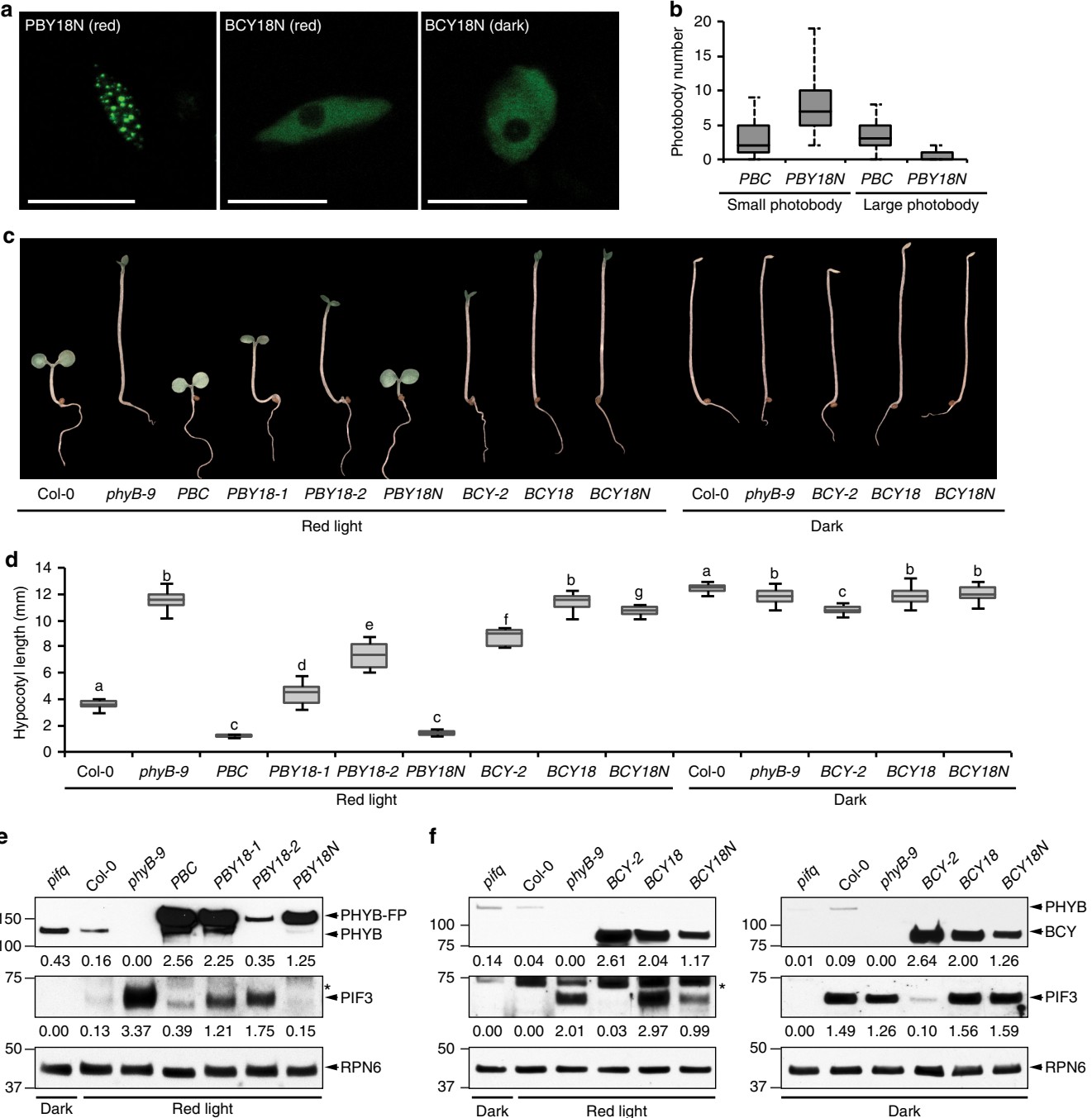

**Fig. 4** Fusing an NLS largely rescues the function of full-length but not the C-terminal module of PHYB18. **a** Confocal images showing the subnuclear localization patterns of PBY18N in 10 μmol m$^{-2}$ s$^{-1}$ R light and BCY18N in red light and darkness. **b** Box and whisker plots showing the number of small (<1 μm$^3$) and large (≥1 μm$^3$) photobodies in hypocotyl epidermal cells of *PBC* and *PBY18N* seedlings grown under 10 μmol m$^{-2}$ s$^{-1}$ R light. **c** Images of representative 4-day-old seedlings of the indicated lines grown under 10 μmol m$^{-2}$ s$^{-1}$ R light or in the dark. **d** Box and whisker plots of hypocotyl measurements of seedlings shown in **c**. The boxes represent from 25th to 75th percentile; the bars equal to the median values. Samples with different letters exhibit statistically significant differences in hypocotyl length (ANOVA, Tukey's HSD, *P* < 0.01, *n* > 40). **e** Immunoblots showing the levels of PHYB and PIF3 in 4-day-old Col-0, *phyB-9*, *PBC*, *PBY18-1*, *PBY18-2*, and *PBY18N* seedlings grown in 10 μmol m$^{-2}$ s$^{-1}$ R light. The sample of dark-grown *pifq* was used as a negative control for the PIF3 Immunoblots. RPN6 was used as a loading control. The relative levels of PHYB, PHYB-FP, and PIF3 were normalized against the corresponding levels of RPN6 and are shown below the blots. The asterisks in the PIF3 blots indicate nonspecific bands. **f** Immunoblots showing the levels of PHYB and PIF3 in 4-day-old Col-0, *phyB-9*, *BCY-2*, *BCY18*, and *BCY18N* grown either in 10 μmol m$^{-2}$ s$^{-1}$ R light or darkness. Dark-grown *pifq* samples were used as negative controls for the PIF3 immunoblots. RPN6 was used as a loading control. The relative levels of PHYB and PIF3 were normalized against the corresponding levels of RPN6 and are shown below each blot

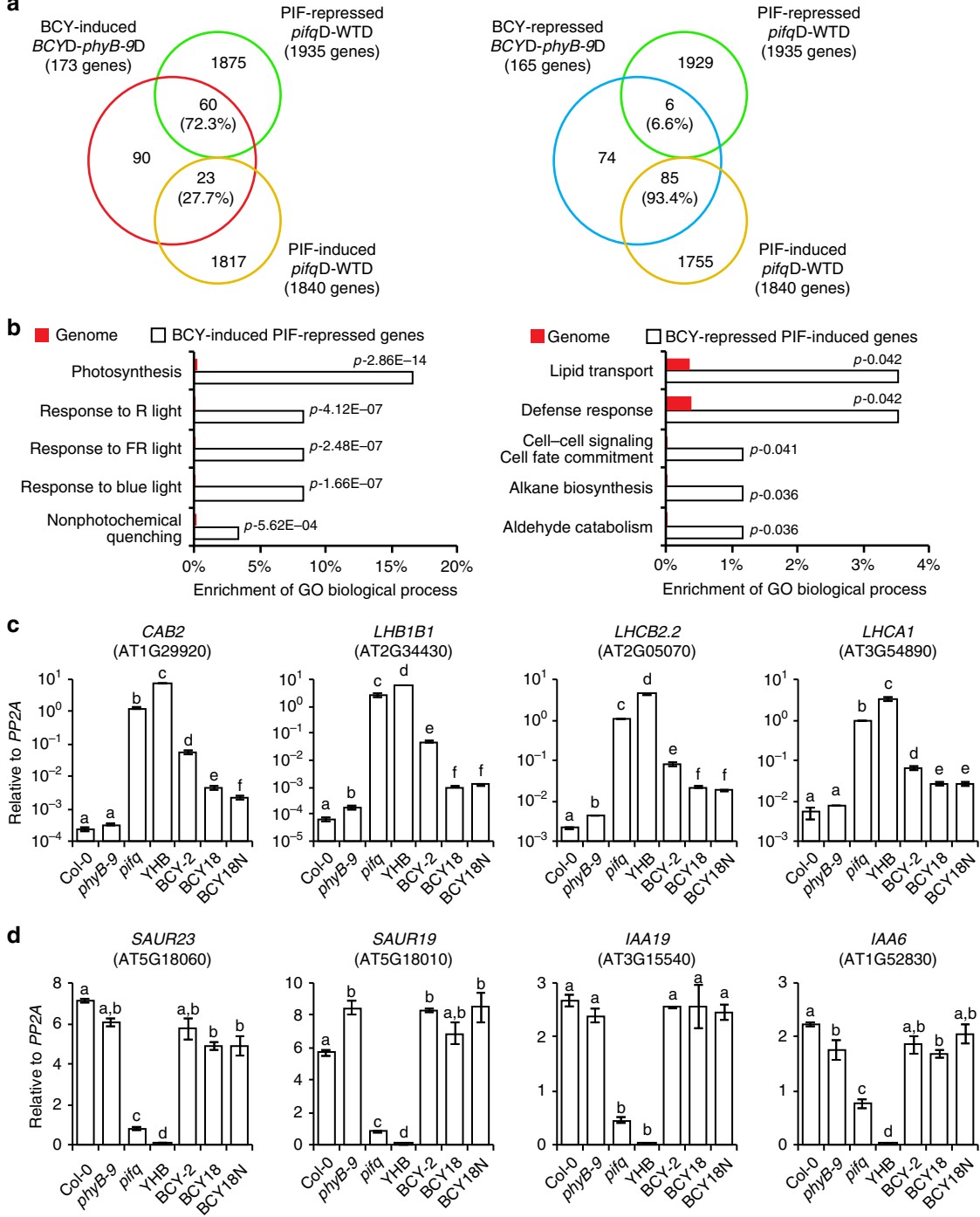

**Fig. 5** The C-terminal module of PHYB regulates a subset of PIF-dependent genes. **a** Venn diagrams showing that 72.3% of BCY-induced and PIF-dependent genes are PIF-repressed and 93.4% of BCY-repressed and PIF-dependent genes are PIF-induced. **b** GO term enrichment analysis showing the top five enriched biological processes with the lowest p-value for the BCY-induced, PIF-repressed genes (left panel) and the BCY-repressed, PIF-induced genes (right panel). **c** qRT-PCR results showing the expression levels of four representative BCY-induced, PIF-repressed photosynthetic genes in 4-day-old Col-0, *phyB-9*, *pifq*, *YHB*, *BCY-2*, *BCY18*, and *BCY18N* seedlings grown in the dark. **d** qRT-PCR results showing the expression levels of four representative PIF-induced growth-relevant genes in 4-day-old Col-0, *phyB-9*, *pifq*, *YHB*, *BCY-2*, *BCY18*, and *BCY18N* seedlings grown in the dark, indicating that this group of growth-related genes are not dependent on BCY

only *BCY-2* and *BCY-3* were slightly shorter than Col-0 and *phyB-9* (Fig. 2e, f), suggesting that BCY is not able to degrade all the PIFs. We therefore tested whether BCY could mediate the degradation of PIF1, PIF4, and PIF5. We used a homemade antibody against PIF1[37] and two commercially available antibodies against PIF4 and PIF5 (Supplementary Fig. 1) to detect the endogenous levels of PIF1, PIF4, and PIF5 in Col-0, *phyB-9*, and the *BCY* lines in continuous R light (Fig. 2d). Interestingly, PIF4 and PIF5, but not PIF1, accumulated to higher levels in *phyB-9* (Fig. 2d), suggesting that PHYB is required for the turnover of PIF4 and PIF5 but not PIF1 in R light. However, in contrast to PIF3, the levels of PIF4 and PIF5 in the *BCY* lines

were similar to those in *phyB-9*, suggesting that the C-terminal module of PHYB is not sufficient to mediate the degradation of PIF4 and PIF5 in R light. To confirm these results, we then examined the levels of PIF1, PIF4, and PIF5 in the dark (Fig. 2g). PIF1 accumulated in all dark-grown *BCY* lines, confirming that BCY does not mediate PIF1 degradation (Fig. 2g). However, the results for PIF4 and PIF5 were surprising. First, PIF4 accumulated fivefold less in *phyB-9* compared with Col-0 (Fig. 2g), suggesting that PHYB is required for the accumulation rather than degradation of PIF4 in the dark. *BCY-1*, *-2*, and *-3* lines rescued the defect in PIF4 accumulation of *phyB-9* (Fig. 2g). PIF5 accumulated to similar levels in Col-0 and *phyB-9*, and the accumulation of PIF5 was also greatly enhanced in the *BCY-1*, *-2*, and *-3* lines (Fig. 2g). These results suggest that BCY can promote the accumulation of PIF4 and PIF5 in the dark. Taken together, these results indicate that the C-terminal module of PHYB can mediate the degradation of only PIF3 but not PIF1, PIF4, and PIF5. We therefore focused on PIF3 degradation in the rest of this study.

**D1040V abolishes activity of the C-terminal module**. To examine whether D1040V affects the activities of PHYB's C-terminal, we generated transgenic lines expressing the C-terminal module of PHYB18 fused to YFP (BCY18). A *BCY18* line expressing a similar level of BCY18 as the level of BCY in *BCY-2* was chosen for further analysis. In contrast to *BCY-2*, *BCY18* showed the same hypocotyl length as *phyB-9* in R light and darkness (Fig. 3a, b). Consistent with the hypocotyl phenotypes, *BCY18* failed to degrade PIF3 in R light and the dark (Fig. 3c, d). We then examined whether D1040V affected the nuclear accumulation of BCY. While BCY was localized to the nucleus in 100% of cells in both R light and darkness, BCY18 was localized to the nucleus in only 63% of cells in R light and 42% of cells in the dark (Fig. 3e). Additionally, in the nuclei with detectable BCY18, nuclear BCY18 failed to localize to photobodies (Fig. 3e). When the nuclear fractions of BCY and BCY18 were compared, the nuclear BCY18 was 3- and 12-fold less than the nuclear BCY in R light and darkness, respectively (Fig. 3f, g). Together, these results demonstrate that D1040V disrupts the function of the C-terminal module of PHYB in nuclear accumulation, photobody localization, and PIF3 degradation.

**Nuclear localization largely rescues PHYB18 but not BCY18**. We then tested whether the phenotype of *phyB-18* is primarily due to the defect of PHYB18 in nuclear accumulation. To that end, we fused an SV40 NLS to the C-termini of PBY18 and BCY18 and named them PBY18N and BCY18N, respectively. We generated transgenic lines expressing either *PBY18N* or *BCY18N* in the *phyB-9* mutant. Both PBY18N and BCY18N were localized to the nucleus but with different photobody localization patterns (Fig. 4a). PBY18N was localized to photobodies in the light (Fig. 4a). However, compared with *PBC*, *PBY18N* had fewer large photobodies and more small photobodies (Fig. 4b), suggesting that PBY18N is defective in the biogenesis of large photobodies. In contrast, BCY18N was evenly dispersed in the nucleoplasm in both R light and dark conditions, indicating that BCY18N is unable to initiate photobody biogenesis. Consistent with the photobody localization patterns, *PBY18N* rescued the long-hypocotyl and PIF3 accumulation phenotypes of *phyB-9*, whereas *BCY18N* was unable to rescue *phyB-9* in either R light or dark conditions (Fig. 4c–f). These results indicate that the main defect caused by the D1040V mutation in full-length PHYB is the defect in nuclear accumulation. However, in the context of the C-terminal module, D1040V blocks its photobody localization and PIF3 degradation in addition to nuclear localization.

**The C-terminus regulates a subset of PIF-regulated genes**. It was surprising that dark-grown *BCY* seedlings accumulated much less PIF3 but more PIF4 and PIF5 (Fig. 2g). To understand how the C-terminal module of PHYB alters the expression of PIF-dependent genes, we compared the global transcriptomic profiles of dark-grown *BCY-2* and *phyB-9* by RNA-seq. These experiments identified 338 genes differentially expressed and statistically significant by 1.5-fold in *BCY-2* compared with *phyB-9* (Supplementary Data 1). Of these 338 BCY-regulated genes, 173 were BCY-induced and 165 BCY-repressed (Fig. 5a and Supplementary Data 1). To determine how many BCY-regulated genes are PIF-dependent, we compared the BCY-regulated genes with a previously defined set of 3775 PIF-regulated genes in 4-day-old seedlings[65]. Among the 173 BCY-induced and 165 BCY-repressed genes, 83 and 91 genes were PIF-dependent, respectively. Interestingly, 72.3% of the BCY-induced/PIF-dependent genes were PIF-repressed genes (Fig. 5a, left panel); 93.4% of BCY-repressed/PIF-dependent genes were PIF-induced (Fig. 5a, right panel). Therefore, the changes in PIF-dependent genes in *BCY-2* reflect the decrease in PIF3. Together, these data provide evidence that the C-terminal module of PHYB alone can alter the expression of a subset of PIF-dependent genes in the dark.

We then performed GO enrichment analysis on the BCY-regulated and PIF-dependent genes. For the BCY-induced/PIF-repressed genes, 13 GO categories were significantly enriched (Supplementary Data 2). The top five GO categories with the lowest *p* values are related to light responses and photosynthesis (Fig. 5b, left panel), particularly genes encoding the light-harvesting apparatus in chloroplasts (Supplementary Data 1). We selected four representative BCY-induced/PIF-repressed genes to verify their expression by quantitative RT-PCR, these include chlorophyll A/B-binding protein 2 (*CAB2*), light-harvesting chlorophyll-protein complexII subunit B1 (*LHB1B1*), photosystem I light-harvesting complex gene1 (*LHCA1*), and photosystem II light-harvesting complex gene 2.2 (*LHCB2.2*). The expression of all four representative genes was indeed highly induced in dark-grown *BCY-2* seedlings, similar to their expression levels in *pifq* and *YHB*—a line expressing constitutively active PHYB[66] (Fig. 5c). In addition, the induction of the photosynthetic genes in the *BCY-2* line is consistent with the previously published transcriptome data of the *pif3* mutant[67–69]. In contrast, the expression of these four genes was less activated in *BCY18* and *BCY18N* (Fig. 5c), indicating that BCY18 and BCY18N were unable to derepress these genes in the dark. These results are consistent with the results that BCY18 and BCY18N failed to degrade PIF3 in the dark (Fig. 4f).

Among the seven enriched GO categories of the BCY-repressed/PIF-induced genes, none was related to auxin (Supplementary Data 2; Fig. 5b, right panel). To further confirm these results, we examined the expression of two auxin signaling genes, *IAA19* and *IAA6*, and two auxin responsive genes, *SAUR23* and *SAUR19*. As shown in Fig. 5d, the expression of these four genes was not altered in *BCY-2*, *BCY18*, and *BCY18N* compared with *phyB-9*. These data are consistent with the published result that knocking out *PIF3* alone has minimum effect on hypocotyl length in the dark[25] and the fact that the *BCY* lines had only minor effects on hypocotyl growth in the dark (Fig. 2e, f).

Taken together, these results show that the C-terminal module of PHYB is sufficient to regulate a subset of PIF-dependent genes. However, we do acknowledge that all differentially expressed genes may not be directly regulated by the decrease level of PIF3. The increased amount of PIF4 and PIF5 may also contribute to these changes. As the nuclear-encoded photosynthetic genes but not the growth-relevant auxin-related genes were regulated by the C-terminal module of

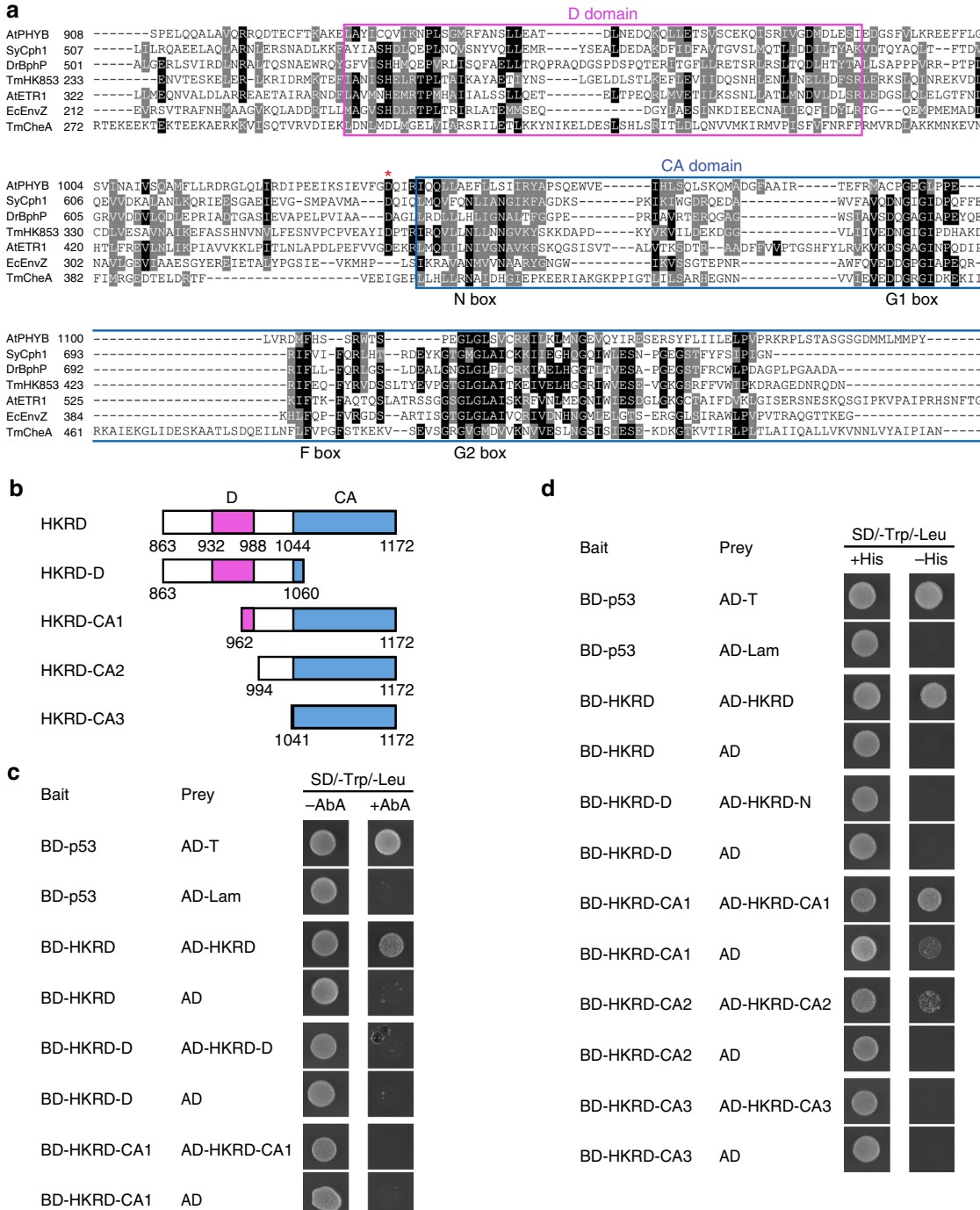

**Fig. 6** The entire HKRD of PHYB is a dimerization domain. **a** Sequence alignment of the HKRD of PHYB with a number of well characterized canonical histidine kinase domains. Identical and similar amino acids are labeled in black and grey background, respectively. The purple box indicates the D domain of PHYB, which is defined based on the DHp domain of EnvZ[80, 81]. The blue box indicates the CA domain, including the conserved N, G1, F, and G2 boxes, which are defined based on the structure of CheA[82]. The red asterisk highlights the D1040 in PHYB. GenBank accession numbers for the selected sequences are: SyCph1, Q55168.1; DrBphP, Q9RZA4.1; TmHK853, Q9WZV7; AtETR1, AAA70047.1; EcEnvZ, P0AEJ4.1; AtphyB, P14713.1; TmCheA, Q56310.2. **b** Schematic illustration of the HKRD fragments used for determining the dimerization domain. The D and CA domains are labeled with magenta and blue boxes, respectively. The numbers indicate the positions of amino acids. **c** Yeast two-hybrid results showing that both the D and CA domains are required for HKRD dimerization. Yeast cells carrying the indicated Bait and Prey constructs were grown on Synthetically-Defined (SD)/-Trp/-Leu selective minimal media with or without 125 ng/ml Aureobasidin A (AbA). Yeast growth in the presence of AbA indicates interaction between the bait and prey proteins. BD, GAL4 DNA-binding domain; AD, GAL4 activation domain; T, SV40 large T antigen; Lam, Lamin. BD-p53/AD-T and BD-p53/AD-Lam are positive and negative controls, respectively. Each bait construct was tested for self-activation by co-transformed with an empty prey vector expressing only AD. **d** Yeast two-hybrid results showing weaker interactions between full-length and fragments of the HKRD on less stringent selective media: SD/-Trp/-Leu or SD/-Trp/-Leu/-His selective media in the absence of AbA. Yeast growth on the SD/-Trp/-Leu/-His plate indicates interaction between the bait and prey proteins. BD GAL4 DNA-binding domain, AD GAL4 activation domain, T SV40 large T antigen, Lam Lamin

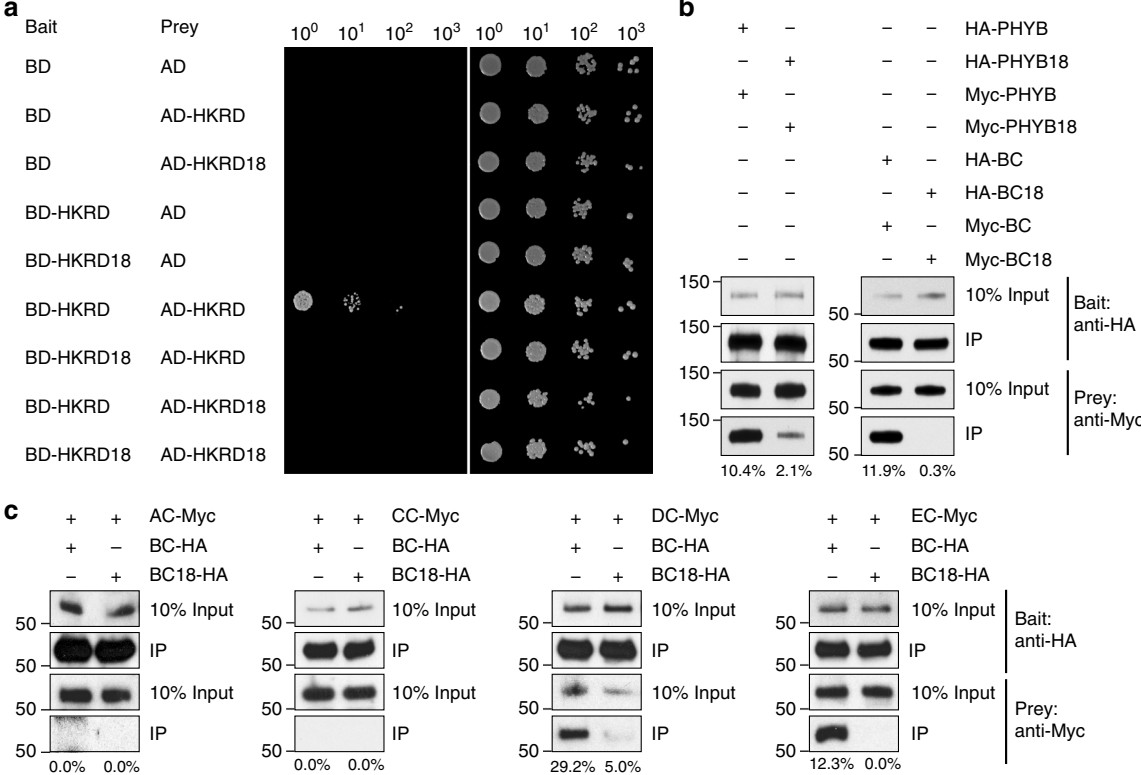

**Fig. 7** D1040V disrupts homo- and heterodimerization of PHYB. **a** Yeast two-hybrid assays showing that the D1040V mutation disrupts the interaction between HKRD. Yeast cells carrying the indicated Bait and Prey constructs were grown on SD/-Trp/-Leu media either with (left) or without (right) AbA. **b** In vitro co-immunoprecipitation results showing that the D1040V mutation disrupts the interaction between the C-terminal domain of PHYB and attenuates the interaction between full-length PHYB. HA-tagged PHYB, PHYB18, PHYB's C-terminal domain (BC), or PHYB18's C-terminal domain (BC18) was co-expressed with the corresponding Myc-tagged version using in vitro transcription/translation and subsequently immunoprecipitated using anti-HA affinity matrix. Bait and prey proteins were detected by immunoblots using anti-HA and anti-Myc antibodies, respectively. The relative amounts of immunoprecipitated Myc-tagged PHYB fragments are shown below the immunoblots. **c** In vitro co-immunoprecipitation results showing that the D1040V mutation in the C-terminal module of PHYB reduces its interaction with the C-terminal modules of PHYD and PHYE. HA-tagged C-terminal module of PHYB (BC) or PHYB18 (BC18) was in vitro co-translated with Myc-tagged C-terminal module of PHYA (AC), PHYC (CC), PHYD (DC) or PHYE (EC) and immunoprecipitated using anti-HA affinity matrix. Bait and prey proteins were detected by immunoblots using anti-HA and anti-Myc antibodies, respectively. The relative amounts of immunoprecipitated Myc-tagged PHY C-terminus are shown below the immunoblots

PHYB, this may explain why the function of the C-terminal module of PHYB had received relatively little attention. This is because inhibition of hypocotyl growth has been used as a standard readout for PHYB activity and the C-terminal module of PHYB alone does not have a major impact on hypocotyl growth (Fig. 2b, e)[53, 54, 63].

**The entire HKRD is a dimerization domain**. The HKRD exhibits high sequence similarity to bacterial histidine kinases[59, 60], which consist of a dimerization and histidine phosphotransfer (DHp) domain and a catalytic and ATP-binding (CA) domain[70]. Because PHYB lacks the conserved histidine in the DHp domain[59, 60], we refer the DHp domain in PHYB as the "D domain" (Fig. 6a). The CA domain of PHYB contain all the conserved N, G1, F, and G2 subdomains for ATP binding[59, 60]. Asp 1040 is conserved among plant, algal, and bacterial PHYs (Supplementary Fig. 2), residing in a linker region between the D and CA domains (Fig. 6a). Therefore, D1040V could disrupt the function of the CA and/or dimerization. Because mutations in conserved residues in the G1 and G2 ATP-binding motifs have little effect on the function of PHYA, it was suggested that ATP-binding of the CA domain is not required for PHY function[71]. Therefore, D1040V might affect a property other than CA activity, one possibility would be dimerization of the HKRD. The HKRD in both PHYA and PHYB was postulated as a

dimerization domain[54–57], but the contribution of D and CA to dimerization has not been determined. To examine whether Asp 1040 is in a region required for dimerization, we examined which region of HKRD is required for dimerization by yeast two-hybrid assays. As predicted, the HKRD interacted with itself in yeast (Fig. 6b, c), confirming that the HKRD contains a dimerization domain. Interestingly, deleting the majority of either CA (HKRD-D) or D (HKRD-CA1) lost the dimerization activity (Fig. 6b, c). The loss of dimerization activity for HKRD-CA1 was surprising, because the same fragment, which contains the C-terminal 210 amino acids, was previously shown to dimerize in yeast[57]. We reasoned that this discrepancy could be due to the high stringency of the yeast two-hybrid assay using the antibiotic aureobasidin A (AbA) as a selection marker. Therefore, we re-examined the interactions by using the HIS3 reporter—a less stringent selection method. The alternative selection method indeed showed that HKRD-CA1 interacted with itself despite its basal self-activation activity (Fig. 6d). In addition, HKRD-CA2, which lacks the entire D domain, also showed weak dimerization activity (Fig. 6b, d). In contrast, HKRD-CA3, which contains the CA domain alone, had no dimerization activity (Fig. 6b, d). Together, these results indicate that the dimerization of HKRD is contributed by the D domain, the CA domain, and the linker region between D and CA, where Asp 1040 resides. Thus, the entire HKRD is a dimerization domain.

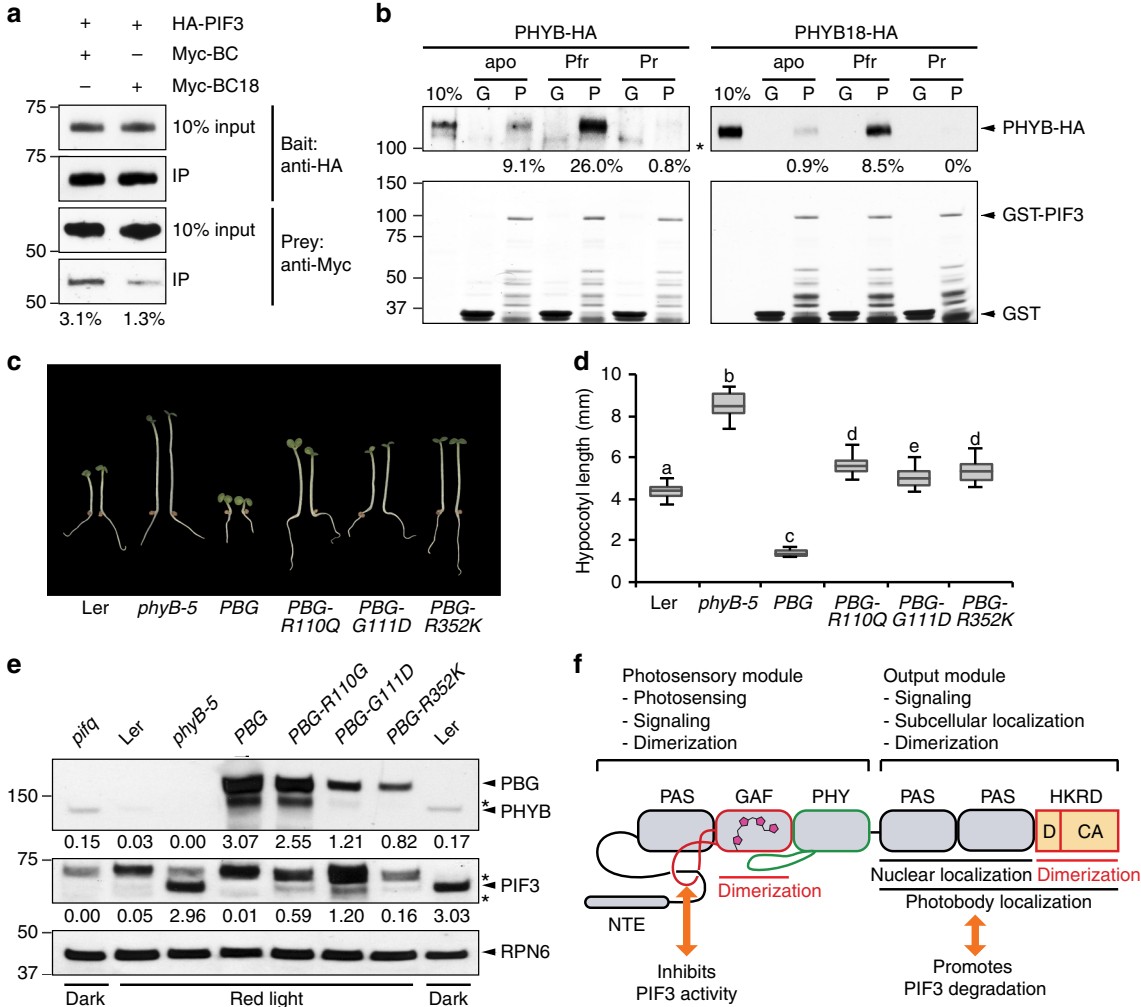

**Fig. 8** The C-terminus of PHYB is a signaling-output module for PIF3 degradation. **a** In vitro co-immunoprecipitation experiments showing that the D1040V mutation disrupts the interaction between the C-terminal module of PHYB (BC) and PIF3. HA-tagged PIFs (HA-PIF3) and Myc-tagged C-terminal domain of PHYB or PHYB18, Myc-BC or Myc-BC18, were co-translated in vitro. HA-PIF3 was pulled down by anti-HA affinity matrix. The input and bound HA-PIF3, Myc-BC, and Myc-BC18 were detected by immunoblots using either anti-HA or anti-Myc antibodies. The estimated bound fractions of Myc-BC and Myc-BC18 are shown below the anti-Myc blots. **b** GST pull-down assays using GST-PIF3 to pull down PHYB-HA and PHYB18-HA in apoprotein, Pr, or Pfr forms. The upper panels are immunoblots using anti-HA antibodies showing bound and input fractions of PHYB-HA or PHYB18-HA. The lower panels are Coomassie Blue-stained gels showing immobilized GST and GST-PIF3. The estimated bound fractions of PHYB-HA or PHYB18-HA for each pulldown assays are shown below the blots. G, GST; P, GST-PIF3. The asterisk indicates nonspecific bands. **c** Images of 4-day-old Ler, *phyB-5*, *PBG*, *PBG-R110Q*, *PBG-G111D*, *PBG-R352K* lines grown in 10 μmol m$^{-2}$ s$^{-1}$ R light. **d** Box and whisker plots of hypocotyl length measurements of the respective seedlings shown in **c**. Samples with different letters indicate statistically significant differences in hypocotyl length (ANOVA, Tukey's HSD, $P < 0.05$, $n > 40$). **e** Immunoblots showing the levels of PHYB and PIF3 in 4-day-old Ler, *phyB-5*, *PBG*, *PBG-R110Q*, *PBG-G111D*, and *PBG-R352K* lines grown in 10 μmol m$^{-2}$ s$^{-1}$ R light. Dark-grown *pifq* samples were used as negative controls for the PIF3 immunoblots. RPN6 was used as a loading control. The relative levels of PHYB and PIF3 were normalized against the corresponding levels of RPN6 and are shown below the blots. Asterisks indicate nonspecific bands. **f** Model for the structure-function relationship of *Arabidopsis* PHYB. The N-terminal photosensory module of PHYB is responsible for light sensing through the bilin chromophore in the GAF domain and subsequent conformational changes of the tongue (green loop). Photoactivation of the N-terminal module induces an Pfr-specific interaction between the knot lasso and PIF3, this interaction contributes to light signaling by repressing the transcriptional activity of PIF3. The GAF domain provides a dimerization interface within the N-terminal module. The C-terminal output module of PHYB interacts directly with PIF3 to mediate its degradation. The PRD is responsible for mediating the nuclear accumulation of PHYB and the entire C-terminal module is required for photobody localization. The entire HKRD provides another dimerization domain. The HKRD communicates with the PRD to facilitate PHYB nuclear accumulation. Dimerization of the HKRD is required for both nuclear and photobody localization of PHYB. The domain structure of PHYB is modified from Burgie and Vierstra[7]

**D1040V disrupts homo and heterodimerization of PHYB**. We then tested whether D1040V could disrupt the dimerization of HKRD. When a D1040V mutation was introduced into either the bait or the prey HKRD, the dimerization activity was abolished (Fig. 7a), supporting the notion that D1040 is involved in the dimerization of HKRD. We then asked whether D1040V could affect dimerization in the context of the C-terminal module and the full-length PHYB. Because the C-terminal module and full-length PHYB have self-activation activity in yeast, we performed pull-down assays instead using in vitro translated PHYB and PHYB's C-terminal module (BC) fused to either HA or Myc tag. As expected, HA-tagged PHYB C-terminal module (HA-BC) was able to pull down Myc-tagged PHYB C-terminal module (Myc-BC) (Fig. 7b), confirming that the C-terminal module of PHYB

interacts directly with itself. In contrast, when the C-terminal module of PHYB18 (BC18) was tested in the pull-down assay, HA-BC18 failed to pull down Myc-BC18 (Fig. 7b), indicating that D1040V disrupts the dimerization of the C-terminal module of PHYB. These data also provide evidence that the HKRD is the only dimerization domain in the C-terminal module of PHYB. In the context of full-length PHYB, although the interaction between HA-PHYB18 and Myc-PHYB18 was reduced by fivefold compared with that between the wild-type PHYB, HA-PHYB18 could still pull down Myc-PHY18 (Fig. 7b), suggesting that although HKRD plays a predominant role in the dimerization of full-length PHYB, there is an additional dimerization domain present in the N-terminal photosensory module. One candidate is the GAF domain, which has been shown to interact directly in the crystallized dimer of PHYB's N-terminal photosenory module[44].

PHYB also heterodimerizes with PHYC, PHYD, and PHYE[57, 72], and the heterodimerization is mediated by the C-terminal module, likely the HKRD[57]. We tested whether D1040V attenuates PHYB heterodimerization by pulldown assays using in vitro co-translated HA-tagged C-terminal module of PHYB or PHYB18 and Myc-tagged C-terminal module of PHYA, PHYC, PHYD, and PHYE. These results show that the C-terminal module of PHYB could pull down PHYD and PHYE but not PHYA and PHYC (Fig. 7c). The D1040V mutation in PHYB dramatically reduced its affinity with PHYD and PHYE (Fig. 7c). Together, these results, combined with the results from previous studies[57], indicate that the HKRD also mediate PHYB heterodimerization and D1040V disrupts PHYB heterodimerization with PHYD and PHYE.

**D1040V attenuates the interaction between PHYB and PIF3**. The interaction between PHYB and PIF3 is a key early signaling event that triggers PIF3 degradation[18, 34, 35, 51, 73]. PIF3 interacts with both the N- and C-terminal modules of PHYB[73]. However, because the interaction between PIF3 and the N-terminal module is stronger and photoconvertible, this interaction has been considered to be a trigger for PIF3 degradation[18, 51, 52], whereas the significance of the interaction between PIF3 and the C-terminal module is less understood. Because the D1040V mutation resides in the C-terminal module, we asked whether D1040V has any impact on its interaction with PIF3. We examined the interaction between HA-PIF3 and the C-terminal module of PHYB and PHYB18 using in vitro immunoprecipitation assays. As shown in Fig. 8a, D1040V reduced the interaction between PIF3 and the C-terminal module by approximately two-thirds. The reduced interaction between PHYB18 and PIF3 may explain the defect in PIF3 degradation in *BCY18N* (Fig. 4f).

We then examined the effect of D1040V on PIF3 binding in the context of the full-length PHYB. These experiments showed that D1040V dramatically reduced the binding of PIF3 to the PHYB apoprotein, Pfr, and Pr forms (Fig. 8b). The interaction between PIF3 and PHYB18-HA was only about one-third of that between PIF3 and PHYB-HA (Fig. 8b). These results suggest that the defect in PIF3 degradation by PHYB18 could be due to its reduced affinity with PIF3. Because D1040V affects the dimerization of the C-terminal module, these results also suggest that dimerization of the HKRD facilitates the PHYB-PIF3 interaction. Together, these results, combined with the results that the C-terminal module is sufficient to mediate PIF3 degradation (Fig. 2) support a new model that PIF3 degradation depends on its interaction with PHYB's C-terminal module.

**PIF3 degradation does not need PHYB N-terminal interaction**. The new hypothesis that PIF3 degradation is dependent on

PHYB's C-terminal module is contradictory to the current model that PIF3 degradation is mediated by the interaction with the N-terminal photosensory module of PHYB[51, 52]. The N-terminal module interacts with PIF3 in a photoreversible manner; PIF3 preferentially interacts with the biologically active Pfr conformer[34, 52]. The residues in the N-terminal module of PHYB involved in the Pfr-specific binding to PIF3 have been identified in the knot lasso[50, 51]. In particular, three mutations, R110Q, G111D, and R352K, were shown to abolish the interaction between PHYB and PIF3 without altering PHYB photoconversion[50, 51]. Moreover, the G111D and R352K mutations also disrupt the binding with other PIFs, including PIF1, PIF4, PIF5, and PIF7[51]. Consistent with their defects in PIF3 binding, these three PHYB mutants were unable to rescue the long-hypocotyl phenotype of *phyB-5*[50, 51]. We reasoned that if the C-terminal module of PHYB was required and sufficient for PIF3 degradation, disrupting the interaction between PIF3 and the N-terminal knot lasso of PHYB should not affect PIF3 degradation. To test this hypothesis, we examined whether the three transgenic lines expressing the full-length PHYB with individual R110Q, G111D, and R352K mutations—*PBG-R110Q*, *PBG-G111D*, and *PBG-R352K*[51]—could still mediate PIF3 degradation. As previously reported, all three lines showed long-hypocotyl phenotypes in R light (Fig. 8c, d) and no hypocotyl phenotypes in the dark (Supplementary Fig. 3a, b). However, in contrast to the long-hypocotyl phenotype, the levels of PIF3 in these lines in R light were greatly reduced compared with *phyB-5* (Fig. 8e), indicating that these *phyB* mutants were still able to mediate PIF3 degradation. As expected, these lines did not affect the PIF3 level in the dark (Supplementary Fig. 3c). These results indicate that PIF3 degradation does not require the interaction between PIF3 and the N-terminal module of PHYB.

## Discussion
A central mechanism in PHY signaling involves direct PHY-PIF interaction and subsequent degradation of PIFs[23]. PIF3 interacts with both the N- and C-terminal modules of PHYB[73]. The widely accepted model indicates that PIF3 degradation is triggered by the light-induced interaction with the N-terminal photosensory module of PHYB through specific residues in the knot lasso[23, 50–52], whereas the C-terminal output module is considered to participate in only subcellular localization and dimerization but not directly in signaling, and the HKRD is thought to be dispensable for PHYB function[53, 63]. In this study, we demonstrate a contrasting light signaling mechanism in which the C-terminal domain of PHYB is the signaling-output module essential and sufficient for mediating PIF3 degradation (Fig. 8f). The light-induced degradation of PIF3 in continuous light does not depend on the photoconvertible interaction between PIF3 and PHYB's N-terminal photosensory module but rather relies on its interaction with the dimeric form of the C-terminal output module. Our results have demonstrated that the HKRD of PHYB acts as a dimerization domain in the C-terminal output module and participates in early light signaling events, including PHYB nuclear accumulation, photobody biogenesis, and PIF3 interaction and degradation. Therefore, similar to bacterial PHYs, the plant phytochrome can also transduce signals through the HKRD not by its histidine kinase activity but rather via dimerization and protein–protein interaction.

Our results indicate that the dimeric form of the C-terminal module of PHYB interacts directly with PIF3 to mediate its degradation (Fig. 8f). This conclusion is supported by the results that the C-terminal module alone was sufficient to trigger PIF3 degradation and activate a distinct set of PIF-repressed

photosynthetic genes (Figs. 2 and 5). We do not believe that this occurred via heterodimerization of the C-terminal domain with another PHY because degradation occurred in the dark when other PHYs would be expected to be inactive and localized outside the nucleus. The signaling function of the C-terminal module depends on its dimer form because the D1040V mutation, which disruptd dimerization of the C-terminal module (Fig. 7a, b), reduced the interaction with PIF3 (Fig. 8a) and abrogated PIF3 degradation in the nucleus (Fig. 4f). The conclusion that PIF3 degradation is mediated by the C-terminal module but not the N-terminal module of PHYB is consistent with the previous results that the N-terminal photosensory module alone is not sufficient for PIF3 degradation[22, 41]. Here, we provide additional evidence that the *phyB* mutants defective in PIF3 binding with the knot lasso of PHYB—R110Q, G111D, and R352K[51]—had little effect on PIF3 degradation (Fig. 8e), demonstrating that PIF3 degradation does not require the photoconvertible interaction with the N-terminal photosensory module of PHYB. Instead, the interaction with the N-terminal module was shown to regulate the transcriptional activity of PIF3[22]. Together, the results presented in this study, combined with previous studies[22, 41], indicate that PIF3 degradation is mediated by the interaction with PHYB's C-terminal output module.

PIF3 binds to the C-terminal module of PHYB relatively weakly compared to the Pfr form of the N-terminal photosensory module[52, 73]. We can not exclude the possibility that in context of full-length PHYB the interaction between PIF3 and the C-terminal module can be enhanced by the light-induced interaction with the N-terminal photosensory module[73]. PIF3 interacts with full-length active PHYB much more strongly than the N- and C-terminal modules separately, suggesting that PIF3 might bind to the two modules cooperatively[73]. However, the PIF3-PHYB interaction does not seem to be regulated by light-induced conformational changes of the C-terminal module per se, because the PHYB-R110Q, PHYB-G111D, and PHYB-R352K mutant proteins, which presumably bind PIF3 through only their C-terminal modules, had similar affinities to PIF3 between their respective Pr and Pfr conformers[51], suggesting that the PIF3 and C-terminal-module interaction is enhanced in the Pfr and therefore not light-dependent. Our results indicate that this weak, non-photoconvertible interaction is capable of mediating PIF3 degradation (Fig. 8f). This conclusion implies that the light-dependence of PIF3 degradation is not specifically due to its light-convertible binding to PHYB but rather is contributed mainly by the light-triggered accessibility of PIF3 in the nucleus through regulation of PHYB's nuclear accumulation. Here, we show that PHYB18, which was defective in nuclear accumulation, was unable to mediate PIF3 degradation (Fig. 1). In addition, fusing an NLS to PHYB18 largely rescued the defects of PHYB18 in mediating PIF3 degradation (Fig. 4). Our results provide additional evidence supporting the notion that PIF3 degradation occurs in the nucleus and that nuclear accumulation of PHYB is a major switch that triggers PIF3 degradation[74].

Our data show that the C-terminal module of PHYB mediates the degradation of PIF3 specifically. *BCY-1* to *-3* lines failed to degrade PIF1 in the dark (Fig. 2g), indicating that BCY does not mediate PIF1 degradation. These results are consistent with the previous findings that PIF1 and PIF3 are degraded through distinct mechanisms. For example, PIF3 is ubiquitylated by the CUL3[LRB] E3 ubiquitin ligase[36], whereas PIF1 is ubiquitylated by the CUL4[COP1-SPA] E3 ubiquitin ligase[75], and COP1 and SPA1 play opposing roles for PIF1 and PIF3 degradation[40, 76]. PHYB also promotes the degradation of PIF4 and PIF5 in continuous R light (Fig. 2d). Our results indicate that the

C-terminal module of PHYB is not sufficient to mediate PIF4 and PIF5 degradation in the light (Fig. 2d). Surprisingly, our results also show that PHYB is required for PIF4 accumulation in the dark and the C-terminal module could promote PIF4 and PIF5 accumulation in the dark (Fig. 2g). The mechanism by which PHYB imposes opposing effects on PIF4 accumulation in the light and dark is still unknown. Taken together, these results provide evidence that the stability of PIFs are differentially regulated via distinct mechanisms.

We further characterized the role of HKRD in dimerization and defined its function in signaling. The HKRD consists of D domain and CA domain (Fig. 6a) and was shown to be a dimerization domain for both homo and heterodimerization[54–57]. Based on its sequence similarity to bacterial histidine kinases[59, 60], the D domain but not the CA domain is predicted to be involved in dimerization. However, our data show that the dimerization of the C-terminal module require both the D and CA domains as well as the linker region between the them (Fig. 6b–d). D1040V disrupted HYB homodimerization and its heterdimerization with PHYD and PHYE (Fig. 7), suggesting that Asp 1040 could be directly involved in dimerization. Heterodimerization of BCY with active PHYD and PHYE could promote their functions in the light and therefore may explain the enhanced activity of the *BCY* line in the light vs darkness (Fig. 2). Supporting this notion, when the D1040V mutation was introduced into the C-terminal module, the *BCY18* line lost the light-dependent discrepancy in the hypocotyl response (Fig. 3a–d). PHYB also heterodimerizes with PHYC[57, 72], but this interaction was not detected by our pulldown assay (Fig. 7c), therefore it is not clear whether Asp 1040 is also involved in the PHYB-PHYC dimerization. In the full-length PHYB, the defect in HKRD dimerization in PHYB18 could be partially compensated by the N-terminal photosensory module (Fig. 7b). Also, PBY18N, but not BCY18N, is active in the nucleus (Fig. 4). These results provide evidence that the N-terminal module facilitates dimerization, likely through the GAF domain (Fig. 8f)[44].

Both the nuclear accumulation and photobody localization of PHYB depend on its C-terminal module[53, 58]. Within the C-terminal module, the PRD is required and sufficient to mediate the translocation of a bulky protein reporter, GUS-YFP, to the nucleus[58]. Although it is still not clear whether the PRD mediates the nuclear import of PHYB by interacting directly with an importin, it is certain that the PRD provides the molecular basis for nuclear accumulation. The previous results suggest that the PRD mediates the nuclear accumulation of PHYB independent of the HKRD. Here, we show that the D1040V mutation in the HKRD abrogates the nuclear accumulation of both full-length and the C-terminal module of PHYB (Figs. 1 and 3), suggesting that the HKRD can communicate with the PRD and modulate its activity in nuclear accumulation, likely via dimerization (Fig. 8f). The HKRD was indicated to regulate the nuclear accumulation of PHYA[77]. The *phyA-402* mutant carries a L946F mutation in the HKRD of PHYA and is defective in PHYA nuclear accumulation[77]. Coincidentally, the L946F mutation lies in PHYA's D domain, it would be interesting to test whether this mutation affects PHYA dimerization.

A more specific mechanism has been proposed for nuclear import of PHYB[78]. This model suggests that PHYB does not contain a bona fide NLS but rather is piggybacked into the nucleus by PIFs[78]. However, the current evidence does not favor this mechanism in planta. As mentioned previously, despite the strong interaction between the N-terminal module of PHYB and PIF3, the N-terminal module alone remains cytoplasmically localized[53, 58]. In addition, the mutations in the knot lasso, R110Q, G111D, and R352K, which disrupt the strong interaction between PIFs and the N-terminal module, had little effect on

PHYB nuclear localization[50, 51]. Our results show that the PHYB18 mutant, which could still interact with PIF3 and presumably other PIFs through the N-terminal module (Fig. 8b), was defective in nuclear accumulation (Fig. 1). Together, these results indicate that binding to PIFs is not sufficient to mediate PHYB nuclear accumulation in *Arabidopsis*, and therefore argue against the PIF-mediated PHYB nuclear important model.

Photobody biogenesis is mediated by the entire C-terminal module of PHYB; neither the PRD nor the HKRD alone is sufficient to mediate photobody biogenesis[17, 53]. Here we show that photobody localization of PHYB requires dimerization of the HKRD (Fig. 4a). In addition, the nuclear-targeted C-terminal module of PHYB18 (BCY18N) was unable to localize to photobodies and to degrade PIF3 in the dark (Fig. 4f), supporting the hypothesis that photobody biogenesis is required for PIF3 degradation[33, 41, 64]. These results are also consistent with the conclusion drawn by mathematical modeling that the PfrPfr dimer of PHYB is required for both photobody localization and the signaling events to inhibit hypocotyl growth[79]. Our results suggest that the constitutively formed photobodies by the C-terminal module of PHYB retain some functions of the regular photobodies in signaling, particularly in the regulation of PIF3 stability.

## Methods

**Plant growth conditions and hypocotyl measurement.** Wild-type Col-0 and L*er*, as well as mutants *phyB-9* (Col-0) and *phyB-5* (L*er*), were used as controls for characterization of light responses. The *PBC* (Col-0) and the *pifq* (Col-0) mutant lines have been previously described[25, 58]. The three *PBG* (L*er*) mutant lines, *PBG-R110Q*, *PBG-G111D*, and *PBG-R352K*, have been previously characterized[50, 51]. All other transgenic lines, including *PBY18*, *BCY*, *BCY18*, *PBY18N*, and *BCY18N*, were generated in the *phyB-9* background and are homozygous for the transgene. Single and triple *pif* mutants, including *pif1-2* (SALK_072677), *pif3-3* (CS66042), *pif4-2* (SAIL_1288_E07), *pif5-3* (SALK_087012), *pif134* (CS66500), *pif135* (CS66047), *pif145* (CS68095), and *pif345* (CS66048) were previously described[25] and obtained from ABRC. *Arabidopsis* seed sterilization and stratification, as well as standard seedling growth experiments in R light and darkness, were performed according to procedures described previously[38]. Fluence rates of light were measured using an Apogee PS200 spectroradiometer (Apogee instruments Inc., Logan, UT). Images of representative seedlings were captured using a Leica MZ FLIII Pursuit Stereo Scope and processed using Adobe Photoshop CC (Adobe Systems, Mountain View, CA). For hypocotyl measurements, 4-day-old seedlings were scanned using an Epson Perfection V700 photo scanner, and hypocotyls were measured using NIH ImageJ software (http://rsb.info.nih.gov/nih-image/). The statistical analyses for hypocotyl length were performed using one-way ANOVA (analysis of variance) with post hoc Tukey's HSD (honestly significant difference) was applied using the online tool at http://statistica.mooo.com/OneWay_Anova_with_TukeyHSD.

**Plasmid construction and generation of transgenic lines.** The primers used for making constructs are listed in Supplementary Table 1. The *PBY18* and *PBY18N* constructs were generated by subcloning full-length or the N-terminal *PHYB* with the D1040V mutation into the KpnI site of pCHF3-YFP and pCHF3-YFP-NLS vectors, respectively. The *BCY18* and *BCY18N* constructs were generated by subcloning the C-terminal sequence of the *PHYB* cDNA (for 594-1172 a.a.) with the D1040V mutation into the KpnI site of pCHF3-YFP and pCHF3-YFP-NLS vectors, respectively. The expression of all the transgenes was driven by the constitutive cauliflower mosaic virus 35S promoter. Transgenic lines were generated by transforming *phyB-9* mutant plants with *Agrobacterium tumefaciens* strain GV3101 harboring the above constructs. For each construct, more than 10 independent T1 lines were selected on half-strength MS medium containing 30 μg/ml kanamycin. Lines that segregated approximately 3:1 for kanamycin-resistance in the T2 generation were selected on the basis of the level of overexpression. T3 self progeny of homozygous T2 plants were used for the experiments.

For making constructs used in yeast two-hybrid, inserts derived from the coding sequence of the wild-type or D1040V mutant of the PHYB HKRD domain (863-1172 a.a.), HKRD-D (863-1060 a.a.), HKRD-CA1 (962-1172 a.a.), HKRD-CA2 (994-1172 a.a.), and HKRD-CA3 (1041-1172 a.a.) were amplified by PCR and ligated into the NdeI and BamHI sites in pGBKT7 or pGADT7AD vectors (Clontech). Constructs for in vitro co-immunoprecipitation assays, including N-terminal Myc- and HA-tagged PHYB, PHYB18, PHYB C-terminus (BC) and PHYB18 C-terminus (BC18), were generated by cloning wild-type or D1040V mutant of PHYB full-length coding sequence or PHYB C-terminal sequence (625-1172 a.a.) into the EcoRI and BamHI sites in pGBKT7 or pGADT7AD vectors using Gibson Assembly (New England BioLabs). Constructs for in vitro co-immunoprecipitation assays, including HA-tagged C-terminal PHYB (BC-HA) and

PHYB18 C-terminus (BC18-HA), and Myc-tagged C-terminal PHYA (AC-Myc), PHYC (CC-Myc), PHYD (DC-Myc), and PHYE (EC-Myc), were generated by cloning wild-type or D1040V mutant of PHYB C-terminal sequence (625-1172 a. a.) into the KpnI and EcoRI sites in the pCMX-PL2-CterHA vector, or by cloning PHYA (563-1122 a.a.), PHYC (553-1111 a.a.), PHYD (598-1164 a.a.), and PHYE (549-1112 a.a.) into the KpnI and EcoRI sites in the pCMX-PL2-CterMyc vector using Gibson Assembly (New England BioLabs). For making constructs used in GST pull-down assays, including PHYB-HA and PHYB18-HA, full-length coding sequence of wild-type or D1040V mutant of PHYB was amplified by PCR and ligated into pCMX-PL2-CterHA between the KpnI and XmaI sites. cDNA prepared from wild-type (Col-0) and *phyB-18* mutant seedlings was used as templates for all the above PCR.

**Total protein extraction and nuclear fractionation.** Total protein was extracted as described previously with minor changes[38]. Briefly, 250 mg 4-day-old seedlings were ground in 750 μl extraction buffer (100 mM Tris-HCl, pH 7.5; 100 mM NaCl; 5 mM EDTA, pH 8.0; 5% SDS; 20% glycerol; 20 mM DTT; 40 mM β-mercaptoethanol; 2 mM PMSF; 1× EDTA-free protease inhibitor cocktail (Roche); 80 μM MG132 (Sigma); 80 μM MG115 (Sigma); 1% phosphatase inhibitor cocktail 3 (Sigma); and 10 mM N-ethylmaleimide) under dim green light. Samples were immediately boiled for 10 min and then centrifuged at 16,000×*g* for 10 min at room temperature. Proteins from the supernatant were used in the subsequent immunoblot assays.

For the fractionation experiments, plant nuclei were isolated from 4-day-old dark- or R-light-grown seedlings as described previously[38] with the following modifications. Tissue was ground to fine powder in liquid N$_2$ and dissolved in a 2× volume of nuclear isolation buffer (20 mM PIPES-KOH, pH 6.5; 2 M hexylene glycol; 10 mM MgCl$_2$; 1× EDTA-free protease inhibitor cocktail (Roche); 0.25% Triton X-100; 5 mM β-mercaptoethanol; 1 mM PMSF). The lysate was filtered through two layers of Miracloth, and the cleared lysate was collected as the "Total" sample. The rest of the filtered lysate was loaded on top of 30% Percoll (Sigma) and centrifuged at 700×*g* for 10 min at 4 °C. The enriched nuclear pellet was dissolved in nuclear isolation buffer and collected as the "nuclear fraction." Both "total" and "nuclear fraction" samples were boiled for 5 min in 1× Laemmli buffer and used in the subsequent immunoblot assays.

**Immunoblots and quantification.** Proteins were separated by SDS-PAGE and blotted onto a nitrocellulose membrane. The membrane was first probed with the indicated primary antibodies, and then incubated with secondary goat anti-rabbit or goat anti-mouse (Bio-Rad) antibodies conjugated with horseradish peroxidase. The signals were detected by a chemiluminescence reaction using the SuperSignal West Pico or Dura Extended Duration Chemiluminescent Substrate (Thermo Fisher Scientific). All immunoblots were repeated at least twice and a representative experiment is shown (Supplementary Fig. 4). Immunoblot bands are quantified using ImageJ software (NIH) and normalized to RPN6 or RNA PolII for nuclear fractions. Mouse monoclonal anti-PHYB (a gift from Dr. Akira Nagatani), rabbit polyclonal anti-PIF1, rabbit polyclonal anti-PIF3, rabbit polyclonal anti-PIF4 (Agrisera, cat. no. AS12 1860), rabbit polyclonal anti-PIF5 (Agrisera, cat. no. AS12 2112), rabbit polyclonal anti-RPN6 (Enzo Life Sciences, cat. no. BML-PW8370-0100), rabbit polyclonal anti-GFP (Abcam, cat. no. ab290), mouse monoclonal anti-RNA Polymerase II (BioLegend, cat. no. 664906), rabbit polyclonal anti-Myc (Thermo Fisher Scientific, cat. no. PA1-981), and goat polyclonal anti-HA (GenScript, cat. no. A00168) antibodies were all used at a 1:1000 dilution.

**Confocal imaging and quantification of photobody morphology.** Four-day-old seedlings grown in continuous R light (10 μmol m$^{-2}$ s$^{-1}$) were fixed in 2% paraformaldehyde in 1× PBS under vacuum on ice for 15 min and then washed with 50 mM NH$_4$Cl in 1× PBS for 2 × 5 min, 1× PBS with 0.2% Triton X-100 for 2 × 5 min, and 1× PBS for 3 × 5 min. Fixed seedlings were mounted with ProLong Gold antifade reagent (Thermo Fisher Scientific), sealed with nail polish, and stored at 4 °C until imaging. Nuclei of hypocotyl epidermal cells were imaged using a Zeiss LSM 510 inverted confocal microscope (Zeiss) with a 100×/1.4 Plan-Apochromat oil-immersion objective. YFP was monitored using 514 nm excitation from an argon laser and a 505–550 nm bandpass detector. CFP was monitored using 458 nm excitation from an argon laser and a 470–500 nm bandpass detector. Images were collected using LSM 510 software version 4.2. and processed using Adobe Photoshop CC software (Adobe Systems). The proportion of cells with or without nuclear signals was manually scored. The volume and number of photobodies were analyzed using Huygens Essential software (Scientific Volume Imaging). The object analyzer tool was used to threshold the image and to calculate the volume of photobodies.

**Yeast two-hybrid assay.** Bait (pGBKT7) and prey (pGADT7AD) vectors described above were transformed into Y2HGold and Y187 yeast strains (Clontech), respectively. Diploid yeast cells were generated by mating single colonies from bait and prey strains and then selected on SD/-Trp/-Leu plates. For Fig. 6, overnight cultures from single yeast colonies were diluted to an OD$_{600}$ of 0.2 and spotted on SD/-Trp/-Leu, SD/-Trp/-Leu/-His, and SD/-Trp/-Leu supplemented with 125 ng/ml Aureobasidin A (AbA). For Fig. 7a, overnight cultures from single

yeast colonies were first diluted to an $OD_{600}$ of 0.2, from which ten-fold serial dilutions were spotted on SD/-Trp/-Leu supplemented with or without 125 ng/ml Aureobasidin A (AbA). The plates were incubated at 30 °C, and pictures were taken on the third day after plating.

**GST pull-down and in vitro co-immunoprecipitation assay.** GST pull-down assays were performed according to the procedure described previously[33]. For the in vitro co-immunoprecipitation experiments, bait and prey proteins were co-expressed using the TNT T7 Coupled Reticulocyte Lysate System (Promega) according to the manufacturer's protocol. The in vitro co-immunoprecipitation assays were performed in E buffer containing 50 mM Tris-HCl, pH 7.5, 100 mM NaCl, 1 mM EDTA, 1 mM EGTA, 1% DMSO, 2 mM DTT, 0.1% Nonidet P-40, and protease inhibitor cocktail (Sigma-Aldrich) as described[33]. Briefly, in vitro translated proteins were incubated with anti-HA Affinity Matrix (Roche) in E buffer at 4°C for 2 h. After incubation, beads were washed four times with E buffer. Bound proteins were eluted by boiling in 1× Laemmli protein sample buffer and subjected to 8% SDS-PAGE. Input and immunoprecipitated proteins were detected by immunoblots using either goat anti-HA polyclonal antibodies (GenScript, cat. no. A00168) or rabbit anti-Myc polyclonal antibodies (Thermo Fisher Scientific, cat. no. PA 1-981).

**RNA extraction and quantitative reverse transcription-PCR.** Total RNA from seedlings of the indicated genotypes was isolated using a Quick-RNA MiniPrep kit with on-column DNase I treatment (Zymo Research). cDNA was synthesized using a Superscript II First-Strand cDNA Synthesis Kit (Thermo Fisher Scientific) according to the manufacturer's recommendations. Oligo(dT) primers (Thermo Fisher Scientific) were used for the analysis of nuclear gene expression. Quantitative RT-PCR was performed with FastStart Universal SYBR Green Master Mix and a LightCycler 96 Real-Time PCR System (Roche). Genes and primer sets used for qRT-PCR are listed in Supplementary Table 2. The statistical analyses were performed using one-way ANOVA (analysis of variance) with post hoc Tukey's HSD (honestly significant difference) was applied using the online tool at http://statistica.mooo.com/OneWay_Anova_with_TukeyHSD.

**RNA-seq.** Total RNA was extracted from 4-day-old dark-grown *phyB-9* and *BCY-2* seedlings using a Quick-RNA MiniPrep kit with on-column DNase I treatment (Zymo Research). Three biological replicates were prepared for each genotype. To make RNA-seq libraries, poly(A) + RNA was first isolated from 1 μg total RNA using the NEBNext Poly(A) mRNA Magnetic Isolation Module (New England BioLabs) and fragmented to an average size of 200 nt. The fragmented and primed poly(A) RNA was immediately used for library constructions using the NEBNext Ultra Directional RNA Library Prep Kit for Illumina (New England BioLabs) according to the manufacturer's instructions.

RNA-seq was performed on an Illumina NextSeq500 at a sequencing depth of ~17.1–21.8 million reads per sample. Raw reads were deposited to the GEO repository under GSE90925. 2 × 75 bp paired-end reads were mapped to the Arabidopsis TAIR10 genome and transcriptome using Tophat/v2.0.14. Gene expression was estimated using Cufflinks/v2.2.1, and then differential gene expression between *BCY* and *phyB-9* was determined by Cuffdiff/v2.2.1 using a *q*-value cutoff of 0.05 and a fold-change equal to or >1.5. Since the RNA-seq libraries were constructed using the NEBNext Ultra Directional RNA library prep kit for Illumina and preserved information about the RNA strand orientation, the tophat and cufflinks parameter "--library-type fr-firststrand" was used to distinguish sense reads and anti-sense reads. Genes with significant expression differences between *BCY-2* and *phyB-9* were further compared.

**Data availability.** The RNA-seq data for dark-grown *phyB-9* and *BCY-2* seedlings have been deposited in GEO repository with the accession code GSE90925. The authors declare that all data supporting the findings of this study are available from the corresponding author upon request.

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

## Acknowledgements

We thank Drs C.Y. Yoo and K. Kim for valuable suggestions regarding the manuscript. This work was supported by grants from NIGMS (R01GM087388) and NSF (IOS-1051602) to M.C. Early stages of these studies were performed in the Chory laboratory, with funding from NIGMS (R01GM052413 to J.C.). J.C. is an investigator of the Howard Hughes Medical Institute.

## Author contributions

Y.Q., E.K.P., J.C. and M.C. conceived the original research plan; M.C. and J.C. supervised the experiments; Y.Q., E.K.P., A.K.R. and M.C. performed the experiments; Y.Q., E.K.P. and M.C. analyzed the data; W.M. analyzed the RNA-seq data; A.N. provided the

transgenic lines expressing *phyB* knot-lasso mutants; Y.Q. and M.C. wrote the article with contributions of all the authors.

## Additional information

**Competing interests:** The authors declare no competing financial interests.

