## [Peer Review File · Nature Communications]

Reviewers' comments:

Reviewer #1 (Remarks to the Author):

The most important contribution of this report is the evidence that the C-terminal PAS-PAS-HKD region of Arabidopsis PHYB (BC) retains partial signaling activity in the absence of light. This conclusion is based on the evidence that expression of a YFP-tagged BC construct (BCY) can effect light-independent changes in expression of a subset of PhANGs (Fig. 5) along with the loss of PIF3 (Fig. 2F, 3D) while triggering only a slight inhibition of hypocotyl growth (Figs. 2D&E, 3A&B and 4C&D) and the evidence that this is due to the loss of selected PIF-dependent growth promotion genes (Fig. 5). Since BCY expression leads to loss of PIF3 (and to the loss of PIF1 and PIF5 by inference), the authors conclude that BCY-mediated PIF turnover is the 'mechanism' responsible for this subset of photomorphogenetic phenotypes in darkness. Most of the rest of the paper is devoted to evidence supporting the second most important conclusion of this paper, i.e. that the strong loss-of-function phenotype of the D1040V mutant of PHYB is due to inhibition of light-mediated nuclear translocation, loss of photobody formation and lack of PIF turnover (Figs. 1, 2, 3, 4 & 7). These measurements convincingly show that the D1040A mutation inhibits light-dependent nuclear localization of full-length PHYB and also inhibits nuclear localization of truncated BCY both in the dark and in the light. These results support the model in Fig. 7D.

On the positive side, this paper provides new evidence supporting multiple signaling roles of the C-terminal domain of phytochromes. This is important because the PHYB CTD was previously shown to be dispensable for hypocotyl growth suppression during early seedling development (ref. 58, 67) - a result that many investigators interpret to mean that the CTD is not important for signal transfer in the nucleus. This region not only plays a critical role in nuclear localization and photobody formation (ref. 60), but also plays an important role in PIF turnover (this work). By contrast, the more well known light-dependent interactions between the phyB NTD and PIFs are necessary and sufficient for light-mediated growth inhibition in the absence of PIF turnover (ref. 21, 72). Thus, this work delimits regions of phyB affecting light-mediated growth suppression (NTD) from those important targeting PIF degradation and PhANG expression (CTD). This work also raises the exciting possibility that the complex between phyB's CTD and PIFs is still able to bind selected PIF targets in the genome to directly regulate gene expression.

On the negative side, the interpretation of the 'light-dependent' phenotypes of BCY expression is complicated by potential heterodimerization with endogenous PHYC, D & E. For example, BCY18 could affect the light-dependent signaling activities of other phytochromes. This concern could be assuaged by additional comparative studies of BCY and PHY lines in a phyABCDE null background. Also missing are controls that directly relate molecular and whole plant phenotypes of dark-grown seedlings with the levels of all relevant PIFs. These include 1) measurements of PIF1, 3, & 5 protein levels in dark-grown BCY, PHY (full length) and phyB-9 seedlings, and 2) comparative phenotypic/molecular analysis of dark-grown BCY seedlings with pif135 triple mutants. These experiments would improve the paper but are not necessarily critical ones for communication of this story.

Clark Lagarias

Reviewer #2 (Remarks to the Author):

In this interesting manuscript, Qiu et al present novel data on the structure function of phytochrome B. They present evidence of the involvement of the C-terminal module in PIF3 degradation, contrary to previous data suggesting that it did not participate directly in signaling. Based on the data shown

using the phyB-18 mutation, they conclude that the C-terminal domain of phyB is sufficient to mediate PIF3 degradation, and that the N-terminal module has little effect on this process. These findings bring together and shed light on existing literature on the structure-function of phyB, and represent an important contribution towards understanding the early light signaling events in phytochrome-mediated responses that will influence thinking in the field. Conclusions are in general convincingly supported by the data. There are, however, some specific points that need further clarification and would benefit from additional evidence to strengthen the conclusions.

-The protein experiments are focused on PIF3 accumulation and degradation. Some of the studied lines (most prominently BCY-2) show almost undetectable levels of PIF3 accumulation in the dark. Is the accumulation of other PIFs affected in a similar manner? Figs 2D-E show that BCY-2 lines are a bit shorter in the dark but otherwise similar to WT, which seems to argue that accumulation of PIF1, 4 and 5 might not be affected. This is important to be able to assess whether or not the described mechanism is restricted to PIF3.

-Along similar lines, transcriptomic data of BCY-induced and repressed genes are compared to pifq-regulated genes. Can authors compare them to transcriptomic data available for individual PIF members? This would allow getting a more complete picture of how the described mechanism affects PIF-regulated signaling.

-In their transcriptomic analysis authors focused on genes that are antagonistically regulated by BCY and PIFs. This analysis needs to be completed and discussed: How many of BCY-induced genes are PIF-induced and how many of the BCY-repressed are PIF-repressed? What is their GO term enrichment?

-Given that the BCY line accumulates lower amounts of PIF protein, it is surprising that not all PIF-regulated genes are affected. How is the specificity for the described gene subsets achieved? How does the observation that PIF-regulated growth-relevant genes are not affected in the BCY lines align with the proposed model?

Minor comments

-The BCY line used for the transcriptomic analysis needs to be indicated in text and Methods.

-It would be useful to include schematic illustrations of the mutants used in each Figure, as it is done in Figure 1. If this is not possible due to figure size constraints, authors might consider including a supplemental figure with illustrations of all constructs used in the manuscript to facilitate the visual reading and interpretation of the data.

Reviewer #3 (Remarks to the Author):

This study has two clearly connected parts. The first part updates the molecular characterisation of a mutation affecting the C-terminal module of phyB and the second part revalidates the importance of the function of the C-terminal module of phyB in downstream signalling. The observation that the C-terminal module of phyB can induce the degradation of PIF3, selected gene expression responses and inhibition of hypocotyl growth is very significant. Particularly because although early mutants had pointed to the C-terminal module of phyB as the signalling module, subsequent experiments had shown that the N-terminal module of phyB without the C-terminal module could induce partial signalling events.

There are, however, some issues that require careful attention:

1) The effect of the C-terminal module of phyB on hypocotyl growth is significantly larger under red light (Fig. 2B) than in darkness (Fig. 2E). In the wild-type, the effect of red light is mediated primarily by phototransformation of phyB from the inactive to the active form (note that the phyB mutant retains a similar length under both conditions). Light activation of phyB requires the chromophore present in the N-terminal domain. The transgenic lines expressing the C-terminal module of phyB have no chromophore attached to phyB. Therefore, the action of the C-terminal module of phyB appears to be enhanced by other phytochrome(s). This could involve a direct interaction and the formation of heterodimers with other phytochrome(s). Another possibility involves a more indirect interaction, where other phytochromes could induce partial decay of PIF3, shifting PIF3 levels to a range where any further decay (caused by the C-terminal module of phyB) is more effective to inhibit growth. However, the level of PIF3 in the phyB mutant is similar to the wild-type in darkness, providing no indication in favour of the latter explanation. This issue has to be addressed in the paper.

2) The authors argue that "the trigger for PIF3 degradation by light is most likely the light-dependent nuclear accumulation of PHYB and photobody biogenesis, by which PIF3 is captured by PHYB for degradation in the nucleus". This could be the case during de-etiolation but it is not clear whether the same model could be applied to the scenario of shade avoidance (a key function of phyB). In the latter case, the seedling are already grown in the light and low red / far-red ratios favour the accumulation of PIF3 but phyB does not leave the nucleus and although the nuclear bodies change their distribution, the presence of PIF3 in these nuclear bodies has not been demonstrated.

3) I do not find information concerning replication of protein blots and statistical treatment of these data. No information is provided about these blots under "Statistical analysis". These data should be treated as any other quantitative data in the paper.

4) The authors argue that "the results presented in this study, combined with previous studies^{21,72}, demonstrate convincingly that the early signalling event of PIF3 degradation is not mediated by the N-terminal photosensory module but rather by the C-terminal output module". I am not convinced of the need of placing one domain above the other. I prefer the more integral view involving the interaction between both domains. Actually, the latter interpretation appears to be postulated in other parts of the text. The differential degree of recovery of the PHYB18 mutation in the full-length and C-terminal domain contexts supports the cooperation of both domains.

Minor issues

5) Figure 7E: Why are the non-specific bands indicated by the asterisks so variable if the loading control is rather homogeneous?

6) I am not sure that the small effect of the C-terminal module of phyB is a good argument to explain why others have not found the effect in similar experiments done in the past. Here all the transgenic lines showed differences with the phyB mutant.

7) The legend to FigS2 is included in the text but not with the figure itself.

Reviewer #1

- 1. On the negative side, the interpretation of the 'light-dependent' phenotypes of BCY expression is complicated by potential heterodimerization with endogenous PHYC, D & E. For example, BCY18 could affect the light-dependent signaling activities of other phytochromes. This concern could be assuaged by additional comparative studies of BCY and PBY lines in a phyABCDE null background.*

Response: BCY-1 to -3 lines showed more pronounced hypocotyl phenotype (Figure 2B, 2C, 2E, and 2F) and PIF3 degradation phenotype (Figure 2D and 2G) in the light compared to darkness. One hypothesis, as suggested by reviewer #1, is that the enhanced activity of BCY in the light is due to its ability in promoting other active PHYs through heterodimerization. PHYB heterodimerizes with PHYC, D, and E (67, 87), and the C-terminal module, likely the HKRD, is involved in heterodimerization (67). Therefore BCY could potentially heterodimerize with full-length activated PHYC, PHYD, PHYE in the light. We performed new in vitro pulldown assays to confirm that PHYB heterodimerizes through its C-terminal module. Although the PHYB/PHYC interaction was not detected by this assay, these experiments confirmed that the

C-terminal module of PHYB interacts with the C-terminal domain of PHYD and PHYE (Figure 7C). More interesting, the D1040V mutation abolished the PHYB/PHYD and PHYB/PHYE interactions, supporting the notion that the HKRD is also involved in heterodimerization of PHYB with PHYD and PHYE. These results also suggest that BCY18 (C-terminal module of PHYB with the D1040V mutation) is unable to heterodimerize with PHYD and PHYE. So, if the enhanced activity of BCY is due to heterodimerization with other active PHYs, BCY18 is expected to lose this light-dependent activity. Indeed, as shown in Figure 3A-D, the *BCY18* line had similar hypocotyl and PIF3 phenotypes in red light and darkness. Taken together, our results suggest that the C-terminal module of PHYB is sufficient to enhance the activity of other active PHYs in the light via heterodimerization, thus providing an explanation for the light-dependent discrepancy in the activity of BCY.

2. *Also missing are controls that directly relate molecular and whole plant phenotypes of dark-grown seedlings with the levels of all relevant PIFs. These include 1) measurements of PIF1, 3, & 5 protein levels in dark-grown BCY, PHY (full length) and phyB-9 seedlings, and 2) comparative phenotypic/molecular analysis of dark-grown BCY seedlings with pif135 triple mutants. These experiments would improve the paper but are not necessarily critical ones for communication of this story.*

Response: We agree that it would be valuable to determine whether the C-terminal module of PHYB can mediate the degradation of other well characterized PIFs - PIF1, PIF4, and PIF5. The main challenge was that antibodies against PIF4 and PIF5 were not available. We first screened and identified commercially-available antibodies against PIF4 and PIF5 (Figure S1). The two commercial PIF4 and PIF5 antibodies and a homemade PIF1 antibody (44) allowed us to examine the protein levels of PIF1, PIF4, PIF5 in Col-0, *phyB-9*, and the *BCY* lines in continuous red light and darkness (Figure 2D, 2G). Collectively, these data show that the C-terminal module of PHYB mediate the degradation of only PIF3.

The degradation of PIF1 in red light is not dependent on PHYB, because PIF1 did not accumulate in *phyB-9* in red light (Figure 2D). In the dark, the *BCY* lines failed to degrade PIF1 (Figure 2G), indicating that the BCY does not mediate PIF1 degradation. These results are consistent with the notion that PIF1 and PIF3 are mediated through different mechanisms with distinct E3 ubiquitin ligases (43, 91).

The accumulation of PIF4 and PIF5 are also negatively regulated by PHYB in the light (Figure 2D). However, the *BCY* lines showed similar levels of PIF4 and PIF5 compared with *phyB-9* in the light (Figure 2D), suggesting that BCY alone is not sufficient to mediate PIF4 and PIF5 degradation. The data regarding PIF4 and PIF5 in the dark were surprising (Figure 2G). Our data show that the accumulation of PIF4, not PIF5, is dependent on PHYB (Figure 2G). BCY alone

was able to promote both PIF4 and PIF5 accumulation in the dark (Figure 2G). Interestingly, PHYB seems to play opposing roles in PIF4 accumulation.

Based on these data, dark-grown *BCY* lines accumulated less PIF3 but more PIF4 and PIF5 in the dark. Interestingly, the global transcriptomic profile of *BCY-2* was more similar to a *pif* mutant - most of the *BCY*-induced genes are PIF-repressed and most of the *BCY*-repressed genes are PIF-induced (Figure 5A). The induction of the nuclear photosynthetic genes is also consistent with the reduced level of PIF3 (Figure 5C). One reason why the growth-related genes were not altered in *BCY* might be due to the accumulation of PIF4 and PIF5 (Figure 5D).

Take together, despite the unexpected and complicated results regarding PIF4 and PIF5 accumulation, the new data show that the C-terminal module of PHYB mediates the degradation of only PIF3, but not PIF1, PIF4, and PIF5. These results provide additional evidence supporting the notion that the stability of different PIFs are regulated by distinct mechanisms.

Reviewer #2:

1. *The protein experiments are focused on PIF3 accumulation and degradation. Some of the studied lines (most prominently BCY-2) show almost undetectable levels of PIF3 accumulation in the dark. Is the accumulation of other PIFs affected in a similar manner? Figs 2D-E show that BCY-2 lines are a bit shorter in the dark but otherwise similar to WT, which seems to argue that accumulation of PIF1, 4 and 5 might not be affected. This is important to be able to assess whether or not the described mechanism is restricted to PIF3.*

Response: Please see the response to the second question of reviewer #1. In summary, the C-terminal module of PHYB mediates specifically the degradation of PIF3, but not that of PIF1, PIF4, and PIF5.

2. *Along similar lines, transcriptomic data of BCY-induced and repressed genes are compared to pifq-regulated genes. Can authors compare them to transcriptomic data available for individual PIF members? This would allow getting a more complete picture of how the described mechanism affects PIF-regulated signaling. In their transcriptomic analysis authors focused on genes that are antagonistically regulated by BCY and PIFs. This analysis needs to be completed and discussed: How many of BCY-induced genes are PIF-induced and how many of the BCY-repressed are PIF-repressed? What is their GO term enrichment?*

Response: The comparison of *BCY*-regulated genes and *pifq*-dependent genes allows us to identify the list of *BCY*-regulated/*PIF*-dependent genes. The reason why we chose to use the *pifq*-dependent genes described in (79) is because their experiments were also performed using 4-d-old seedlings. The transcriptome analyses of individual and combinations of *pif* mutants

were mostly performed using 2-d-old seedlings. Our preliminary data showed that data from 2-d-old seedlings had less overlapping with our dataset. Therefore, we did not go further to compare our dataset with those for each individual *pif* mutants. We have revised Figure 5A to show the percentage of BCY-dependent genes that are either induced or repressed by PIFs. The new data show that 72% of the BCY-induced/PIF-dependent genes were PIF-repressed and 93% of the BCY-repressed/PIF-dependent genes were PIF-induced (Figure 5A). Therefore, although *BCY-2* had reduced amount of PIF3 and increased amount of PIF4 and PIF5 (Figure 2G), the *BCY-2* line behaved like a *pif* mutant as opposed to a PIF overexpression line. The GO term enrichments of all of the four categories of BCY-regulated/PIF-dependent genes are listed in Dataset 2.

3. *Given that the BCY line accumulates lower amounts of PIF protein, it is surprising that not all PIF-regulated genes are affected. How is the specificity for the described gene subsets achieved? How does the observation that PIF-regulated growth-relevant genes are not affected in the BCY lines align with the proposed model?*

Response: The new data provided an explanation - in the *BCY* lines, although the amount of PIF3 was reduced and the levels of PIF4 and PIF5 were increased. The enhanced levels of PIF4 and PIF5 could explain why the growth-relevant genes were not affected or still activated. The activation of the photosynthetic genes might imply that their repression in the dark is particularly sensitive to the level of PIF3.

Minor comments

1. *The BCY line used for the transcriptomic analysis needs to be indicated in text and Methods.*

Response: We added the line “*BCY-2*” in the text and Methods.

2. *It would be useful to include schematic illustrations of the mutants used in each Figure, as it is done in Figure 1. If this is not possible due to figure size constraints, authors might consider including a supplemental figure with illustrations of all constructs used in the manuscript to facilitate the visual reading and interpretation of the data.*

Response: The transgenic lines expressed either full-length or the C-terminal domain of PHYB. The domain structure of full-length PHYB is shown in Figure 1A. We included a new schematic for the C-terminal module of PHYB (*BCY*) in Figure 2A.

Reviewer #3:

1. *The effect of the C-terminal module of phyB on hypocotyl growth is significantly larger under red light (Fig. 2B) than in darkness (Fig. 2E). In the wild-type, the effect of red light is mediated primarily by phototransformation of phyB from the inactive to the active form (note that the phyB mutant retains a similar length under both conditions). Light activation of phyB requires the chromophore present in the N-terminal domain. The transgenic lines expressing the C-terminal module of phyB have no chromophore attached to phyB. Therefore, the action of the C-terminal module of phyB appears to be enhanced by other phytochrome(s). This could involve a direct interaction and the formation of heterodimers with other phytochrome(s). Another possibility involves a more indirect interaction, where other phytochromes could induce partial decay of PIF3, shifting PIF3 levels to a range where any further decay (caused by the C-terminal module of phyB) is more effective to inhibit growth. However, the level of PIF3 in the phyB mutant is similar to the wild-type in darkness, providing no indication in favour of the latter explanation. This issue has to be addressed in the paper.*

Response: Please see the response to the first question of Reviewer #1. Our results supports the first hypothesis that the enhanced activity of the C-terminal module of PHYB (BCY) is due to heterodimerization with other active PHYs, such as PHYD and PHYE.

2. *The authors argue that “the trigger for PIF3 degradation by light is most likely the light-dependent nuclear accumulation of PHYB and photobody biogenesis, by which PIF3 is captured by PHYB for degradation in the nucleus”. This could be the case during de-etiolation but it is not clear whether the same model could be applied to the scenario of shade avoidance (a key function of phyB). In the latter case, the seedling are already grown in the light and low red / far-red ratios favour the accumulation of PIF3 but phyB does not leave the nucleus and although the nuclear bodies change their distribution, the presence of PIF3 in these nuclear bodies has not been demonstrated.*

Response: The same model also applies to shade conditions, because there is no fundamental difference in terms of the light-dependent mechanisms of PHYB nuclear accumulation, photobody biogenesis, and PIF3 degradation under shade vs other light conditions. Photobody localization of PHYB is dependent on the red light intensity or the percentage of Pfr (19), or more accurately based on the mathematical model, it depends on the percentage of PHYB in the dimeric PfrPfr form (98). Similar to dim red light, shade will reduce the amount of PfrPfr of PHYB and therefore will reduce photobody localization of PHYB (see Figure 1A and 1C of reference 19). These previously published data show that the steady-state morphology of photobody respond to light quantity and quality. In addition, photobody biogenesis correlates with PIF3 degradation. For example, during light-to-dark or light-to-FR-pulse, re-accumulation of PIF3 is tightly correlated with the disappearance of photobodies (48). Moreover, PIF3 and PHYB have been shown to colocalize to photobodies before PIF3 degradation during the dark-to-light transition (20, 47). Together, these data support the proposed model. Although under certain shade conditions, considerable amount of PHYB remain in the nucleus, the decrease in the PfrPfr level will reduce photobody localization of PHYB as well as PHYB/PIF3 interaction and therefore leads to PIF3 accumulation.

3. *I do not find information concerning replication of protein blots and statistical treatment of these data. No information is provided about these blots under “Statistical analysis”. These data should be treated as any other quantitative data in the paper.*

Response: All immunoblots have been repeated at least twice and most by three times or more. A representative experiment was shown. This information is in Methods under Immunoblot.

4. *The authors argue that “the results presented in this study, combined with previous studies^{21,72}, demonstrate convincingly that the early signalling event of PIF3 degradation is not mediated by the N-terminal photosensory module but rather by the C-terminal output module”. I am not convinced of the need of placing one domain above the other. I prefer the more integral view involving the interaction between both domains. Actually, the latter interpretation appears to be postulated in other parts of the text. The differential degree of recovery of the PHYB18 mutation in the full-length and C-terminal domain contexts supports the cooperation of both domains.*

Response: We feel that our results, combined with previously published data, strongly support the conclusion. Our results show that the C-terminal module alone is sufficient to mediate PIF3 degradation. Previous data show that the N-terminal module of PHYB alone cannot mediate PIF3 degradation (24, 48), despite its activity in repressing PIF3 activity (24). Here, we further demonstrate the interactions between PIF3 and N-terminal module of PHYB are not required for PIF3 degradation (Figure 8). Together, these data strongly argue that the N-terminal domain of PHYB does not participate directly in PIF3 degradation. It is still possible that the N-terminal is involved in the degradation of other PIFs, which remains to be tested. The PHYB18 data was intended to explain that the N-terminal domain could indirectly contribute to PIF3 degradation by providing a weaker dimerization domain through the GAF.

Minor issues

1. *Figure 7E: Why are the non-specific bands indicated by the asterisks so variable if the loading control is rather homogeneous?*

Response: We recognized these bands as nonspecific bands based on the controls using *pifq* and a combination of *pif* mutants (Figure S1). We also noticed that some nonspecific bands were variable but don't have a good explanation. It is possible that the amounts of the nonspecific proteins are dependent on certain genotype or light conditions.

2. *I am not sure that the small effect of the C-terminal module of phyB is a good argument to explain why others have not found the effect in similar experiments done in the past. Here all the transgenic lines showed differences with the phyB mutant.*

Response: Although the hypocotyl phenotypes of the *BCY* lines were significant, *BCY* had an overall minor effect on hypocotyl growth. This is also consistent with the global transcriptomic profile where the growth-relevant genes were not altered (Figure 5D). Looking back to the literature, for example, in Figure 1b of Matsushita et al (63), the hypocotyl length of the line “CG” expressing the C-terminal module of PHYB was slightly shorter than the *phyB* mutant control. It was likely that this minor reduction in hypocotyl length was considered as “inactive” for PHYB function and was thus ignored. Indeed, without examining the level of PIF3, the activity of the C-terminal module of PHYB could never be predicted.

3. *The legend to FigS2 is included in the text but not with the figure itself.*

Response: We included the legend in the figure.

Reviewers' comments:

Reviewer #1 (Remarks to the Author):

The authors have nicely addressed all of my concerns. This is a very important communication.

Reviewer #2 (Remarks to the Author):

Authors show new data to address and determine that the C-terminal module of phyB mediates the degradation of PIF3 only. Surprisingly, they describe that BCY alone is not sufficient to mediate PIF4/5 degradation in the light and is able to promote PIF4/5 accumulation in the dark. This reviewer acknowledges all the additional work done, which supports the interesting notion that different PIFs are regulated by distinct mechanisms. However, I still have some reservations about the transcriptomic data in Figure 5. Authors argue that "BCY-2 behaves like a pif mutant rather than a PIF-ox line" and also that "One reason why the growth-related genes were not altered in BCY might be due to the accumulation of PIF4 and PIF5". These explanations seem contradictory to me. I think it may be even more relevant now to compare BCY-regulated genes to transcriptomic data available for individual PIF factors. I agree that seedling age can have some effect, but the overall picture should be supportive of your model. And there is actually a microarray published in 4d-old dark-grown pif3 seedlings (Sentandreu et al., 2011), the same conditions used in this work for phyB-9 and BCY-2. Additionally, authors can look at the expression of selected genes in 5C and 5D in pif single mutants. I believe adding these data to the manuscript will strengthen the conclusions.

Response to Reviewers

Editor and Reviewer #3

Reviewer #3 made comments to the editor only. Given your response to point 1 raised by reviewer #3 and point 1 of reviewer #1, reviewer #3 felt that it was not possible at present to rule out that the effects of the C-terminal domain demonstrated in the present manuscript could be mediated by interaction with other phytochromes and could implicate the N-terminal domain of these phytochromes in signaling. Therefore referee #3 felt that it was not possible to confidently conclude that the C-terminal domain of phyB was sufficient for mediating PIF3 degradation. We have discussed this concern editorially and we do not feel that it precludes further consideration of the manuscript. However, we would like to consider your response to this concern before making a final decision. While we appreciate that you do provide data showing that the interaction between the N-terminal of phyB and PIF3 is not required for light induced PIF3 degradation, it is not clear to us editorially, whether or not it is possible to rule out the involvement of the N-terminal of other phytochromes in this situation, and we do still feel that any uncertainty on this point should be addressed in a revision. I would be happy to discuss this point with you further by e-mail or over the phone prior to resubmission if you feel it would be helpful.

Response:

We disagree with reviewer #3 on the comment that “it was not possible to confidently conclude that the C-terminal domain of phyB was sufficient for mediating PIF3 degradation”. Our results show that expressing the C-terminal module of PHYB alone was able to degrade PIF3 in the dark, where the rest of the phytochromes were in the inactive Pr form (Figure 2G). These results demonstrate clearly that the C-terminal module of PHYB is sufficient for mediating PIF3 degradation.

Maybe our response to the reviewers’ questions confused the reviewer because we did not separate the hypocotyl response from PHYB-mediated PIF3 degradation. In the continuous red light, PIF3 degradation requires PHYB only, as knocking-out PHYB alone leads to PIF3 accumulation (Figure 2D, Qiu et al. 2015 *Plant Cell* 27:1409-27). Here we show that the C-terminal module of PHYB is sufficient to mediate PIF3 degradation in the dark. These results are consistent with the published results demonstrating that the N-terminal module alone cannot degrade PIF3 in the light (Park et al. 2012 *Plant J* 72:537-46; Van Buskirk et al. 2014 *Plant Physiol* 165:595-607). This study further demonstrates that particularly the interaction between PIF3 and the N-terminal module of PHYB is not required for PIF3 degradation (Figure 8E), whereas the interaction between PIF3 and the C-terminal module is needed for PIF3 degradation by the C-terminal module of PHYB (Figure 8A-B). Together, these results show convincingly that the C-terminal module of PHYB plays a signaling output role by interacting directly with PIF3 to mediate its degradation.

The role of PHYB's C-terminal module in regulating hypocotyl growth is unlikely through PIF3 but likely through other PIFs. First, expressing the C-terminal module of PHYB is not effective in inhibiting hypocotyl growth in the dark (Figure 2E-F). Although the BCY-2 line could mediate PIF3 degradation in the dark, it affected only the photosynthetic genes but not the prototypic growth-relevant genes (Figure 5C-D). These observations are consistent with the published results that the *pif3* mutant had minimal effect on hypocotyl growth in the dark (Leivar et al. 2008 *Curr Biol* 18:1815-23). In our response to point 1 by reviewer #3, we tried to explain how the activity of BCY in regulating hypocotyl growth was enhanced in the light (Figure 2B, 2C, 2E, 2F). One possibility is that this enhanced hypocotyl inhibition activity is provided by other phytochromes through heterodimerization with the C-terminal module of PHYB. We agree that in this case, the N-terminal module of the other phytochromes could play a role in inhibiting hypocotyl growth. However, this extra activity is most likely through the inhibition of the activity of the other PIFs, similar to the mechanism by which the N-terminal module of PHYB inhibits the activity of PIF3 (Park et al. 2012 *Plant J* 72:537-46). We did not point out this mechanism of hypocotyl inhibition by the other phytochromes in our previous response, which might cause confusion. Nonetheless, regardless how the heterodimers of BCY and other phytochromes work in the light, our data of PIF3 degradation by the C-terminal module of PHYB in the dark, in combination with data published previously and presented here, demonstrate convincingly that PIF3 degradation is mediated by the C-terminal module of PHYB.

Reviewer #2

Authors show new data to address and determine that the C-terminal module of phyB mediates the degradation of PIF3 only. Surprisingly, they describe that BCY alone is not sufficient to mediate PIF4/5 degradation in the light and is able to promote PIF4/5 accumulation in the dark. This reviewer acknowledges all the additional work done, which supports the interesting notion that different PIFs are regulated by distinct mechanisms. However, I still have some reservations about the transcriptomic data in Figure 5. Authors argue that "BCY-2 behaves like a pif mutant rather than a PIF-ox line" and also that "One reason why the growth-related genes were not altered in BCY might be due to the accumulation of PIF4 and PIF5". These explanations seem contradictory to me. I think it may be even more relevant now to compare BCY-regulated genes to transcriptomic data available for individual PIF factors. I agree that seedling age can have some effect, but the overall picture should be supportive of your model. And there is actually a microarray published in 4d-old dark-grown pif3 seedlings (Sentandreu et al., 2011), the same conditions used in this work for phyB-9 and BCY-2. Additionally, authors can look at the expression of selected genes in 5C and 5D in pif single mutants. I believe adding these data to the manuscript will strengthen the conclusions.

Previous studies have shown that PIFs collectively repress the nuclear-encoded photosynthetic genes and induce the prototypical growth-relevant genes in the dark (Leivar et al. 2009 *Plant Cell* 21:3535-53; Zhang et al. 2013 *Plos Genet* 9:e1003244). Because *BCY-2* had less amount of PIF3 but increased levels of PIF4 and PIF5 in the dark, it was not predictable as for

how the PIF-dependent genes would change. Our RNA-Seq data show that the *BCY-2* line de-repressed the expression of photosynthetic genes without significant alterations of the prototypical growth-relevant genes (Figure 5). These results suggest that the transcriptional profile of *BCY-2* is more similar to the *pif3* mutant. To confirm this point, as suggested by reviewer #2, we compared our dataset with the published transcriptome data of the *pif3* mutant. The transcriptome of dark-grown *pif3* has been reported three times before - Sentandreu et al. 2011 *Plant Cell* 23:3974-91, Zhang et al. 2013 *Plos Genet* 9:e1003244, and Pfeiffer et al. 2014 *Mol Plant* 7:1598-618. Because these studies were carried out under different growth conditions and seedling ages, as expected, these three datasets had only a few overlapping genes (attached supplemental Figure X), indicating that the transcriptome data vary largely from different experimental setups. It is important to point out that although the experiments by Sentandreu et al. were also performed using 4-day old seedlings, they used the pseudo-dark condition where a residual amount of active PHYB still existed, whereas our experiments were conducted under the true-dark condition without active PHYB (Leivar et al. 2008 *Curr Biol* 18:1815-23). As a result, it is expected that our dataset should be different from the Sentandreu data as well. Nonetheless, we compared our data to the published datasets individually (supplementary Figure X). The results show that photosynthetic genes are enriched in the BCY-induced, PIF3 repressed genes. These results support the notion that the transcriptome of the *BCY-2* line reflects the downregulation of PIF3. But, with the poor consistency of the transcriptome data, we are inclined not to include the data analysis in the manuscript. Instead, we revised the manuscript by citing the above three references and stating that the *BCY-2* transcriptome data are similar the *pif3* mutant with regard to the upregulation of the photosynthetic genes and minimum change of growth-relevant genes and hypocotyl growth.

Figure X. BCY-induced and PIF3-repressed genes enrich photosynthetic genes. (A) The three published datasets of PIF3-dependent genes have only a small number of overlapping genes. Venn diagram showing the overlapping genes among the three published datasets of PIF3-dependent genes. (B) Comparisons between BCY-induced genes and the individual published datasets of PIF3-repressed genes show that the overlapping genes are enriched with photosynthetic genes. These results support the notion that the upregulation of photosynthetic genes in *BCY-2* is likely due to PIF3 degradation.